# Long-term intravital imaging of the multicolor-coded tumor microenvironment during combination immunotherapy

**Shuhong Qi**[1,2†], **Hui Li**[1,2†], **Lisen Lu**[1,2], **Zhongyang Qi**[1,2], **Lei Liu**[1,2], **Lu Chen**[1,2,3], **Guanxin Shen**[3], **Ling Fu**[1,2], **Qingming Luo**[1,2*], **Zhihong Zhang**[1,2*]

[1]Britton Chance Center for Biomedical Photonics, Wuhan National Laboratory for Optoelectronics-Huazhong University of Science and Technology, Wuhan, China; [2]MoE Key Laboratory for Biomedical Photonics, Department of Biomedical Engineering, Huazhong University of Science and Technology, Wuhan, China; [3]Department of Immunology, Tongji Medical College, Huazhong University of Science and Technology, Wuhan, China

**\*For correspondence:** qluo@mail. hust.edu.cn (QL); czyzzh@mail. hust.edu.cn (ZZ)

[†]These authors contributed equally to this work

**Competing interests:** The authors declare that no competing interests exist.

**Abstract** The combined-immunotherapy of adoptive cell therapy (ACT) and cyclophosphamide (CTX) is one of the most efficient treatments for melanoma patients. However, no synergistic effects of CTX and ACT on the spatio-temporal dynamics of immunocytes in vivo have been described. Here, we visualized key cell events in immunotherapy-elicited immunoreactions in a multicolor-coded tumor microenvironment, and then established an optimal strategy of metronomic combined-immunotherapy to enhance anti-tumor efficacy. Intravital imaging data indicated that regulatory T cells formed an 'immunosuppressive ring' around a solid tumor. The CTX-ACT combined-treatment elicited synergistic immunoreactions in tumor areas, which included relieving the immune suppression, triggering the transient activation of endogenous tumor-infiltrating immunocytes, increasing the accumulation of adoptive cytotoxic T lymphocytes, and accelerating the infiltration of dendritic cells. These insights into the spatio-temporal dynamics of immunocytes are beneficial for optimizing immunotherapy and provide new approaches for elucidating the mechanisms underlying the involvement of immunocytes in cancer immunotherapy.

## Introduction

Cancer immunotherapy has been an area of significant process in recent decades, and is considered a breakthrough in cancer therapy (*Couzin-Frankel, 2013*). Current promising techniques in cancer immunotherapy are the use of monoclonal antibodies (mAbs), tumor vaccines, and adoptive cell therapy (ACT) (*Elert, 2013*; *Couzin-Frankel and McNutt, 2013*). ACT has been reported to be one of the most efficient treatments for patients with melanoma because it effectively elicits anti-tumor immune responses and leads to objective tumor regression in more than 50% of patients (*Pittet et al., 2007*; *Rosenberg et al., 2008*). During the immunotherapy-elicited immune response, various immune cells, including endogenous and adoptive immunocytes, are activated in the tumor microenvironment (*Schreiber et al., 2011*; *Gajewski et al., 2013*). However, the underlying process remains a 'black box' because, whereas information is available to describe the input (immunotherapy) and the output (tumor elimination), relatively limited information is available to describe the spatio-temporal changes in the participating immune cells in the tumor microenvironment (*Krummel, 2010*). Traditional biological methods, such as immunohistochemistry methods or flow

**eLife digest** Melanoma is a form of skin cancer that is particularly difficult to treat. A new approach that has shown a lot of promise in treating many different cancers, including melanoma, is called "immunotherapy". This technique harnesses the immune system – the body's natural defences that help to protect against infections and disease – to combat cancer.

One powerful type of immunotherapy involves injecting patients with cells called lymphocytes, which form part of the immune system. This is known as adoptive cell therapy and can activate the immune system to fight cancer, helping to shrink tumors. This treatment can be made even more powerful by combining it with a drug called cyclophosphamide and this combination, known as CTX-ACT, is currently one of the most efficient treatments for melanoma. Yet, little information is available to indicate why this treatment is so effective.

Using mice implanted with melanoma cells, Qi, Li et al. sought to understand how CTX-ACT treatment works, with the goal of optimising it to increase its success. The results showed that a protective barrier of immune cells that suppresses the anti-tumor immune response – called an "immunosuppressive ring" – surrounds untreated tumors. CTX-ACT treatment can breakdown these rings, helping to reactivate the anti-tumor immune reaction in the tumors. This allows both the injected and mouse's own immune cells to move into the tumor and destroy cancer cells.

Qi, Li et al. used their findings to optimise treatment and succeeded in controlling tumor growth in the mice for several weeks. These new insights could be used to improve current immunotherapies, and offer new approaches for investigating the involvement of immune cells in the treatment of a wide range of different cancers.

cytometry, have revealed the type and function of immune cells that play a role at certain time-points during the anti-tumor reaction (*Perentes et al., 2009*; *Boissonnas et al., 2013*). Visualization of the dynamic characteristics of immunocytes, including their movement, migration, and recruitment, as well as of the interaction between immunocytes and tumor cells in the systemic environment, is critical for understanding the success or failure of cancer immunotherapy on a mechanistic level (*Krummel, 2010*; *Bourzac, 2013*).

The common strategy used in clinical trials of ACT is to isolate tumor antigen-specific lymphocytes from patients, robustly expand these cells in vitro, and infuse them into cancer patients (*Yee et al., 2002*; *Mackensen et al., 2006*; *Benlalam et al., 2007*). The activity and cytotoxicity of tumor-specific cytotoxic T lymphocytes (CTLs) have long been considered the crucial factors for the efficacy of ACT against solid tumors (*Boon, 1994*). However, recent reports have indicated that despite the detection of tumor-reactive CTLs in the bloodstream after the adoptive transfer, most CTLs lost their anti-tumor functions as a result of endogenous immunosuppressive networks in the tumor microenvironment (*Zippelius et al., 2004*; *Baitsch et al., 2011*). Foxp3$^+$ regulatory T cells (Tregs) play a key immunosuppressive role in the endogenous immunosuppressive networks (*Curiel et al., 2004*; *Vignali et al., 2008*). Various immunosuppressive mechanisms associated with Tregs have been revealed, including their specific recruitment and accumulation in the tumor stroma (*Wang et al., 2004*; *Nishikawa and Sakaguchi, 2014*), suppression of the CTL function by TGF-β (*Chen et al., 2005*), and induction of the CTL dysfunction by interactions of Tregs with intratumoral dendritic cells (DCs) (*Bauer et al., 2014*). However, the details of the spatio-temporal orchestration of Tregs associated with tumor progression, the contributions of the Treg distribution in tumor stroma to immunosuppressive networks, and the effect of Tregs on the infiltration of adoptive CTLs into solid tumors remain unclear.

Treg elimination is considered to be an essential component for disrupting the immunosuppressive networks in the tumor microenvironment before ACT (*Rosenberg et al., 2008*; *Antony et al., 2005*). Cyclophosphamide (CTX) is a commonly used alkylating agent in chemotherapy, which has been applied in recent years to selectively suppress, abrogate and deplete Tregs before ACT treatment (*Rosenberg et al., 2008*; *Nishikawa and Sakaguchi, 2014*; *Oleinika et al., 2013*; *Zhao et al., 2010*). Lymphodepletion before ACT by CTX treatment further promoted tumor-specific CTL infiltration into the target tumors (*Bracci et al., 2007*; *Rosenberg et al., 2008*; *Boissonnas et al., 2013*).

The migratory behavior of effector T cells in solid tumors has been analyzed by two-photon micros-copy, proving that migration presents the critical step for the sufficient infiltration of T cells into the solid tumor and promotes tumor elimination (*Mrass et al., 2006*; *Boissonnas et al., 2007*). However, the influence of CTX treatment on the long-term precise migratory behavior of adoptive CTLs in solid tumors has not been directly observed in vivo.

In addition to the selective depletion and suppression of Tregs, other immunomodulatory mecha-nisms of CTX treatment include the systemic activation of DCs (*Radojcic et al., 2010*; *Veltman et al., 2010*; *Nakahara et al., 2010*), increase of the infiltration of DCs into the tumor area (*Schiavoni et al., 2011*), and promotion of the penetration of CTLs into the tumor parenchyma (*Boissonnas et al., 2013*; *Bracci et al., 2007*). Although certain immunomodulatory functions of CTX alone or in combination with immunotherapy have been proposed previously, the synergistically enhanced effects of the combined CTX and ACT (CTX-ACT) treatment on the adoptive and endoge-nous anti-tumor immunoreactions have not been revealed. In particular, the dynamic effects of the CTX-ACT treatment on endogenous immune cells (such as CTLs, neutrophils and DCs) in the tumor microenvironment have not been well described. Using intravital microscopy and a skin-fold window chamber model, we captured a long-term comprehensive sequence of cellular events associated with multicolor-coded tumor cells, adoptive CTLs, endogenous CTLs, Tregs, DCs and tumor-infiltrat-ing immunocytes in the tumor microenvironment during CTX-ACT combined-immunotherapy for B16 melanoma. The long-term intravital imaging data provided direct evidence for the synergetic anti-tumor immune-enhanced effects of the CTX-ACT combination treatment, including the elimina-tion of Tregs, activation of the endogenous tumor-infiltrating immunocytes, and infiltration of adop-tive and endogenous CTLs and DCs.

## Results

### Limited effect of adoptive CTLs on B16 tumor cells in vivo

To assess the curative efficacy of ACT on murine melanoma in vivo, CTLs were obtained from the splenocytes of C57BL/6 mice immunized with mitomycin C pre-treated B16 melanoma cells accord-ing to previous protocols (*Restifo and Nicholas, 2011*) and reports (*Liu et al., 2006*; *Bauer et al., 2014*). Here, the activity and cytotoxicity of CTLs against target tumor cells were demonstrated in vitro by flow cytometry of B16 tumor cells that stably express mutants of the cyan fluorescent pro-tein mCerulean (CFP-B16 tumor cells, *Figure 1—figure supplement 1*), which is minimally immuno-genic in C57BL/6 mice (*Skelton et al., 2001*; *Yang et al., 2016*). However, the ACT treatment failed to control the tumor growth (*Figure 1A*), which is consistent with the results reported for certain clinical cases (*Zippelius et al., 2004*; *Mukai et al., 1999*).

To determine the reasons for the failure of adoptive CTLs, we detected the distribution of these cells in different organs of CFP-B16 tumor-bearing mice at an early stage (1–3 days after the adop-tive transfer of CTLs) and a later stage (4–6 days after the adoptive transfer of CTLs). Adoptive CTLs stained with the fluorescent dye CFSE (5-(and-6)-carboxyfluorescein diacetate succinimidyl ester, CFDA SE) were intravenously transferred into tumor-bearing mice on the sixth day after the implantation of tumor cells, which we defined as Day 0. The distribution of the CTLs was assessed in vivo by fluorescence imaging of frozen tissue sections from the naïve lymph nodes (NLNs), tumor-draining lymph nodes (TDLNs), spleens and tumors. The data indicated that the CTLs primarily accu-mulated in the TDLNs but rarely accumulated in the tumors during either the early or the late stage (*Figure 1B*, *Figure 1—figure supplement 2*).

### Tregs formed an 'immunosuppressive ring' around the solid tumor

To visualize the immune reaction in the tumor microenvironment under immunotherapy dynamically, we established a murine model of the multicolor-coded tumor microenvironment using fluorescent protein (FP) transgenic C57BL/6 mice with a skin-fold window chamber implanted with CFP-B16 tumor cells. Here, several types of FP transgenic C57BL/6 mice were used to represent the host immunocytes: Foxp3[+] regulatory T cells were labeled with red FP (mRFP), endogenous tumor-infil-trating lymphocytes were labeled with green FP (GFP), CD11c[+] dendritic cells were labeled with yel-low FP (YFP), and tumor-infiltrating immunocytes (TIIs) were labeled with enhanced green FP (EGFP).

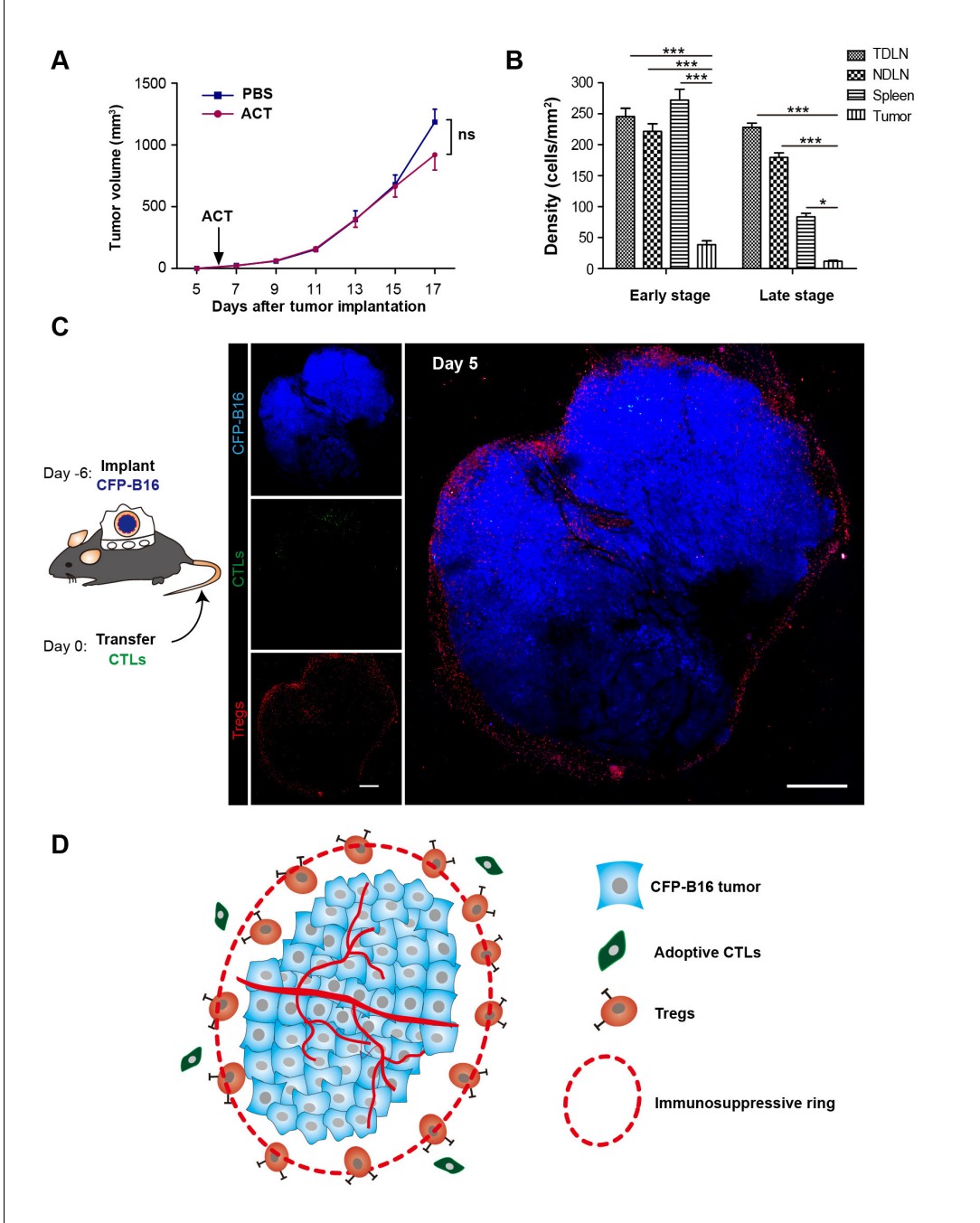

**Figure 1.** 'Immunosuppressive ring' formed by Tregs in CFP-B16 tumor-bearing mice, which inhibited the anti-tumor efficacy of the adoptive CTLs. (**A**) Tumor growth curves for CFP-B16 tumors of mice treated with ACT or PBS control. The data are represented as the mean ± SEM tumor volume (n = 12–14, three independent experiments). ns: not significant, (***Figure 1—source data 1***). (**B**) Density of carboxyfluorescein succinimidyl ester (CFSE)-labeled CTLs within different organs (tumor-draining lymph nodes (TDLNs), non-tumor-draining lymph nodes (NDLNs), spleens, and tumors). The density was determined by counting the number of CFSE-labeled CTLs per mm$^2$ on frozen tissue sections. The data are represented as the mean ± SEM (n = 18–22 fields, 0.18 mm$^2$ per field) from three independent experiments. *p<0.05, **p<0.01, ***p<0.001, (***Figure 1—source data 2***). (**C**) Large-field intravital images of an 'immunosuppressive ring' around the CFP-B16 tumor. Blue – CFP-B16 tumor; red – Tregs (*Foxp3*-mRFP cells); green – CFSE-labeled CTLs. The left panel shows different single color channels of the tumor microenvironment, and the right panel shows the three color channels merged. Scale bar: 500 μm. (**D**) Schematic diagram of the 'immunosuppressive ring' in the tumor microenvironment.

The following source data and figure supplements are available for figure 1:

**Source data 1.** Tumor growth curves for CFP-B16 tumors of mice treated with ACT or PBS.

*Figure 1 continued on next page*

*Figure 1 continued*

**Source data 2.** Density of CFSE-labeled CTLs within different organs.

**Figure supplement 1.** Establishment and characterization of CFP-B16 tumor-specific CTLs.

**Figure supplement 2.** Fluorescence microscopy images showing the distribution of CFSE-labeled CTLs in different organs.

**Figure supplement 3.** Long-term and large-field imaging of the process by which Tregs formed an immunosuppressive ring.

The intravital imaging data revealed that only a few CFSE-labeled adoptive CTLs infiltrated into the tumor area, regardless of an early or late stage post-adoptive CTL transfer (*Figure 1C*, *Figure 1—figure supplement 3*). Compared with the rare infiltration of adoptive CTLs, Tregs (*Foxp3*-mRFP cells) were clearly observed infiltrating the tumor microenvironment on Day 1, before they gradually accumulated at the edge of the tumor to form a ring on Day 5 (*Figure 1C*, *Figure 1—figure supplement 3*). This observation suggested that an 'immunosuppressive ring' formed by Tregs surrounded the solid tumor to prevent the recruitment of adoptive CTLs, as illustrated in the schematic diagram in *Figure 1D*.

## CTX treatment eliminates Tregs, blocks the formation of the immunosuppressive ring, and accelerates the infiltration of adoptive CTLs into the tumor

To verify the immunosuppressive effect of Tregs on the infiltration of adoptive CTLs into the tumor areas, CTX (150 mg kg$^{-1}$) was used to specifically abrogate Tregs on the fourth day after the implantation of CFP-B16 tumor cells. This CTX dose was selected on the basis of CFP-B16 tumor inhibition experiments with different CTX doses in immune-competent C57BL/6 mice and immune-deficient BALB/c nude mice. The results of these experiments showed that, compared with other CTX doses (50 mg kg$^{-1}$ and 100 mg kg$^{-1}$) usually used in tumor immunotherapy, only the dose of 150 mg kg$^{-1}$ CTX controlled the CFP-B16 tumor growth successfully when CTX was used alone as well as when it was used in combination with ACT (*Figure 2—figure supplement 1A*). Furthermore, a tumor growth inhibition experiment performed on the BALB/c nude mice, confirmed that the effective anti-tumor effect of CTX at the dose of 150 mg kg$^{-1}$ is not due to its direct cytotoxicity, because no difference in the inhibition of tumor growth was observed between the PBS control group and the mice treated with different doses of CTX (50, 100 or 150 mg kg$^{-1}$, *Figure 2—figure supplement 1B*). These results suggest that the efficacy of the 150 mg kg$^{-1}$ CTX treatment for CFP-B16 tumor in C57BL/6 mice depends on the enhanced anti-tumor immune response, and not on the direct cytotoxicity of the treatment.

Two days later, in-vitro-expanded and activated CTLs were intravenously transferred into the tumor-bearing mice; this day was defined as Day 0 (six days after tumor cells implantation). The CTX-ACT combined treatment significantly inhibited tumor growth compared with the ACT treatment alone, but produced results that were not significantly different from those produced by the CTX treatment alone (*Figure 2A*). The spatio-temporal changes in immunocytes in the tumor microenvironment were monitored on Days 1–7 using large-field confocal microscopy.

First, we assessed whether the CTX treatment effectively depleted Tregs. Intravital imaging (*Figure 2B*) demonstrated that without CTX treatment, the margin of the tumor was surrounded by a layer of Tregs on Day 5 (*Figure 2B*, top left corner). The CTX treatment successfully eliminated most of the Tregs in the tumor area and removed the 'immunosuppressive ring' formed by Tregs (*Figure 2B*, bottom left corner). These imaging results were further confirmed by immunohistochemical staining using the anti-Foxp3 antibody (*Figure 2B*, right panel). In addition, the number of CFSE-labeled CTLs in the tumor area was clearly increased in CTX-ACT-treated mice, compared with that in mice received the ACT treatment alone (*Figure 2B*, left panel, *Figure 2—figure supplement 2*).

Next, long-term large-field intravital imaging revealed the entire immune reaction in the tumor microenvironment after CTX-ACT treatment, from the accumulation of adoptive CTLs to the

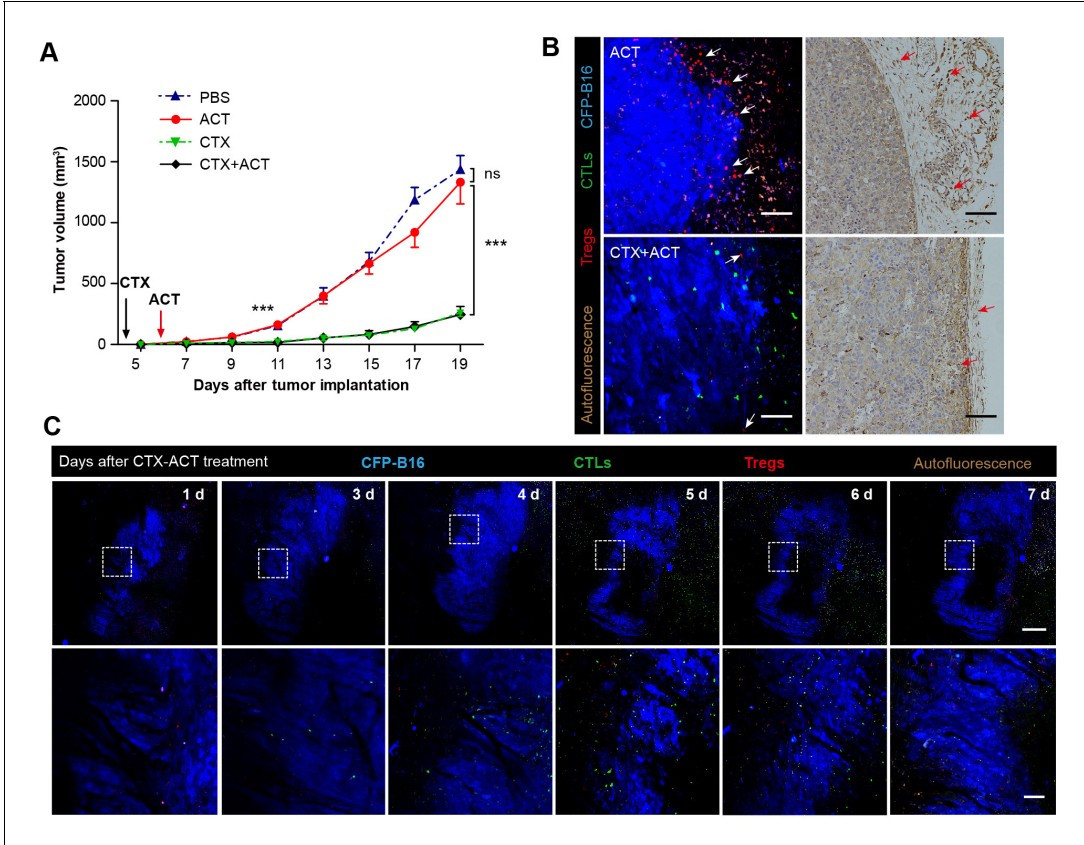

**Figure 2.** Synergistic effect of CTX and the adoptive CTLs (ACT) on CFP-B16 tumor immunotherapy. (**A**) Growth curves for the CFP-B16 tumors treated with ACT, CTX or CTX-ACT and the PBS control. The data are represented as the mean ± SEM tumor volume (n = 12–14, three independent experiments). ns: not significant, ***p<0.001, (*Figure 2—source data 1*). (**B**) Intravital confocal fluorescence imaging of Tregs (red) at the tumor periphery (left panel), and immunohistochemistry (right panel) of the CFP-B16 tumor tissues after ACT (top row) or CTX-ACT (bottom row) treatment. Scale bars: 100 μm. The arrows indicate Tregs. (**C**) Long-term intravital imaging of the multicolor-coded tumor environment in CTX-ACT-treated mice. Red – Tregs (Foxp3-mRFP); green –CSFE-labeled CTLs; blue –CFP-B16 tumor. Top row: large-field images; scale bar: 500 μm. Bottom row: images from the region of interest in the top row; scale bar: 100 μm. The imaging data are representative of similar results from 3–5 mice in two independent experiments.

The following source data and figure supplements are available for figure 2:

**Source data 1.** Growth curves for the CFP-B16 tumors treated with ACT, CTX, CTX-ACT or PBS control.

**Figure supplement 1.** Evaluation of the effect of different doses of CTX and CTX combined with ACT on CFP-B16 tumor growth in vivo.

**Figure supplement 1—source data 1.** Growth curves for the CFP-B16 tumors in C57BL/6 and BALB/c nude mice treated with different doses of CTX, different doses of CTX combined with ACT treatment and PBS control.

**Figure supplement 2.** Quantification of the intravital imaging of Tregs and adoptive CTLs in ACT- and CTX-ACT-treated mice.

**Figure supplement 2—source data 1.** Density of Tregs and adoptive CTLs in the tumor area.

destruction of the tumor (*Figure 2C*). These images showed that in the combined treatment, few adoptive CTLs infiltrated into the tumor microenvironment during the early stage (Days 1–3). Subsequently, the accumulation of CTLs increased significantly during the late stage (Days 4–6, *Figure 2— figure supplement 2*). By contrast, mRFP-Tregs were rarely observed regardless of the respective stage (*Figure 2—figure supplement 2*). The elimination of Tregs and the infiltration of CTLs resulted in large-scale tumor death consisting of tumors 'shrinking' from the outside and 'melting' from the

inside (*Figure 2C*, Days 5–6). This finding suggested that the infiltrating CTLs destroyed the solid tumor by applying two approaches: attacking the tumor cells at the periphery according to an 'out-side-in' pattern and eliminating tumor cells at the parenchyma according to an 'inside-out' pattern. The anti-tumor effect of the adoptive CTLs peaked at Days 5–6. One day later (*Figure 2C*, Day 7), the number of adoptive CTLs decreased rapidly in the tumor area because of the death or dysfunction of CTLs, leading to tumor regrowth.

## Four stages of adoptive CTLs migration in the tumor microenvironment after CTX-ACT treatment

The migration of adoptive CTLs is the crucial step in accelerating the penetration of CTLs into the solid tumors. To understand the migratory behavior of the adoptive CTLs in the tumor microenvironment, we obtained dynamic information about the adoptive CTLs using confocal laser scanning microscopy (CLSM) from Day 1 to Day 6 after the CTX-ACT treatment (*Figure 3A–D*). Subsequently, we quantified the motility of the adoptive CTLs in detail over four stages. Three parameters were used to describe the motility properties of the CTLs in vivo (*Figure 3F–H*): the mean velocity, which represents the migratory speed; the confinement ratio, which indicates the ratio of the maximum displacement of each cell from its path length within a given time (*Boissonnas et al., 2007*; *Cahalan and Parker, 2008*); and the arrest coefficient, which denotes the percentage of time that each cell remained arrested (*Boissonnas et al., 2007*).

The first stage corresponds to Days 1–2. During this stage, a number of adoptive CTLs migrated to the tumor periphery (*Figure 3A*, Day 1) with a mean velocity of $3.23 \pm 2.18$ µm min$^{-1}$ (n = 80 cells from 4–5 mice, *Figure 3F*) and only a few CTLs infiltrated into the tumor parenchyma (*Video 1*).

The second stage corresponds to Days 3–4. During this stage, adoptive CTLs accumulated at the periphery of the tumor (mean velocity – $3.13 \pm 2.43$ µm min$^{-1}$, confinement ratio – $0.57 \pm 0.29$, and arrest coefficient – $48 \pm 36\%$, n = 305 cells; *Figure 3A,F–H*, Day 3) and their motility was similar to that at Day 1. Nevertheless, a few of adoptive CTLs were able to infiltrate into the tumor parenchyma, and remained confined with constrained trajectories (*Figure 3C,D*) and slow speed (mean velocity – $0.87 \pm 0.28$ µm min$^{-1}$, confinement ratio – $0.32 \pm 0.21$, and arrest coefficient – $91 \pm 8\%$; n = 118 cells; *Figure 3F–H*), suggesting that these CTLs formed stable interactions with neighboring tumor cells (*Video 2*).

During the third stage (Day 5), a larger number of adoptive CTLs migrated to the tumor periphery and their mean velocity dramatically increased to $5.79 \pm 3.12$ µm min$^{-1}$ (n = 428 cells), which was much faster than the mean velocities calculated during the first two stages (p<0.001; *Figure 3F*). Consistently, the migration trajectories at this stage were less confined (confinement ratio: $0.67 \pm 0.26$; *Figure 3B,G*), and the arrest coefficient decreased to $22 \pm 30\%$ (*Figure 3H*). The CTLs in the deep tumor parenchyma also displayed a trend toward greater speed with a marked increase in their mean velocity to $3.22 \pm 2.58$ µm min$^{-1}$ (n = 193 cells; *Figure 3F*), less confined trajectories (confinement ratio – $0.60 \pm 0.25$; *Figure 3B,G*) and a decrease in the arrest coefficient ($50 \pm 39\%$; *Figure 3H*). The changes in the migratory behavior of the CTLs in both tumor periphery and parenchyma were consistent with the observed tumor destruction (*Figures 2C* and *3C*, *Video 3*), indicating that when most tumor cells were killed, the adoptive CTLs resumed their high-speed movement to search for new targets.

During the fourth stage (Day 6), the motility of CTLs near the tumor periphery remained high (mean velocity – $5.89 \pm 3.9$ µm min$^{-1}$, confinement ratio – $0.69 \pm 0.24$, and arrest coefficient – $26 \pm 34\%$; n = 147 cells; *Figure 3F–H*). By contrast, the motility of CTLs in the parenchyma significantly decreased as evidenced by their more confined trajectories and an obvious increase in the arrest coefficient (mean velocity – $0.86 \pm 1.49$ µm min$^{-1}$, confinement ratio – $0.46 \pm 0.22$, and arrest coefficient – $88 \pm 29\%$; *Figure 3F–H*). The decreased number and motility of the CTLs and tumor regrowth suggested that the adoptive CTLs were dysfunctional or died during this stage (*Video 4*). The mean displacement analysis revealed that all of the adoptive CTLs in both the tumor periphery and the tumor parenchyma displayed random walking during different stages (*Figure 3E*).

These data suggested that the migratory behavior of adoptive CTLs in the tumor areas at different stages is correlated to their anti-tumor efficacy. The dysfunction and death of adoptive CTLs and tumor regrowth during the fourth stage suggested that a subsequent round of CTX-ACT combined treatment is required to maintain the anti-tumor immune activation.

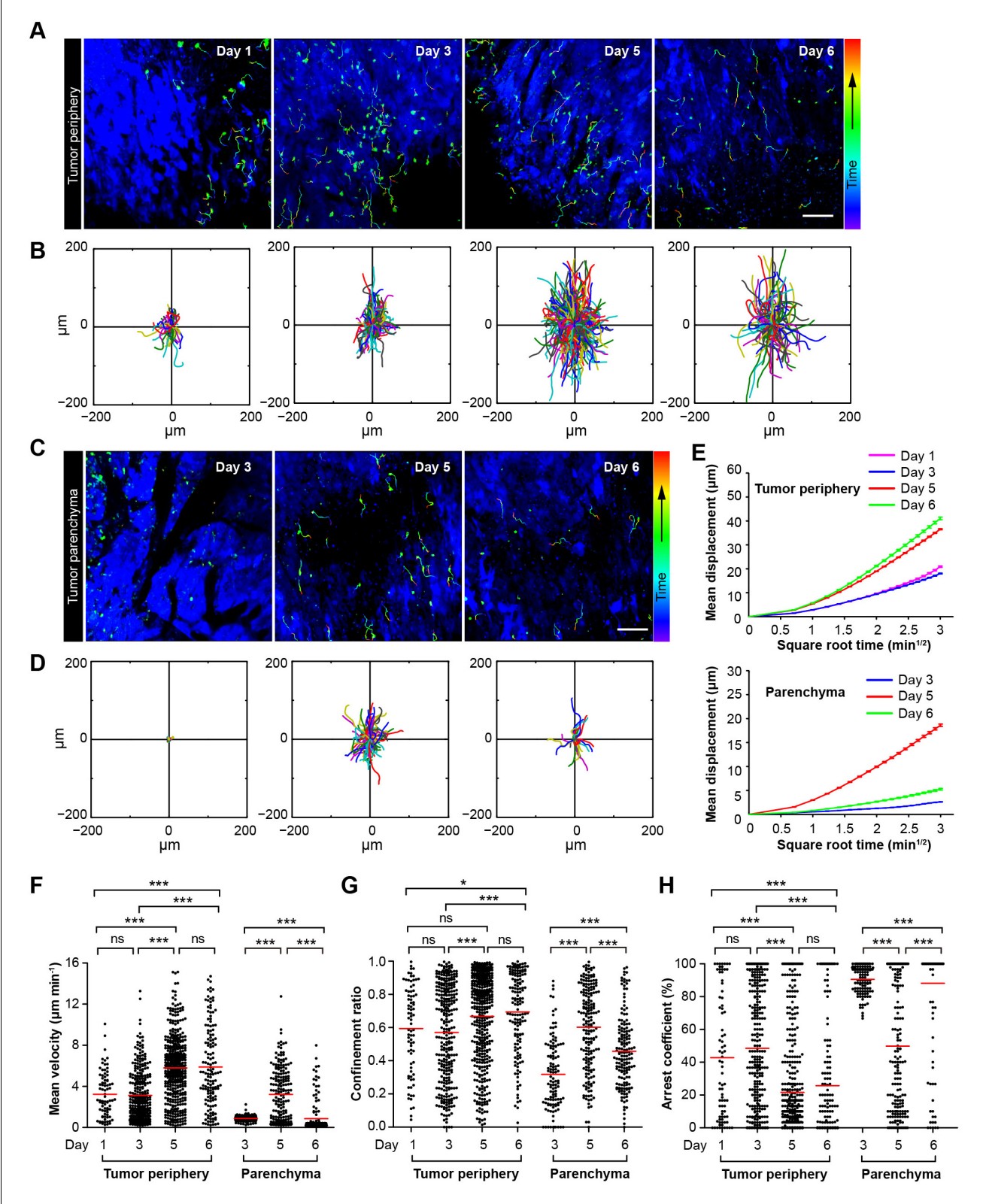

**Figure 3.** Migratory behavior of the adoptive CTLs in the tumor microenvironment of mice treated with CTX-ACT. (A–D) Time-lapse images of CTLs with time-coded motion trajectories (color scale represents the duration). (A,C) Images of CTLs (green) at the periphery (near the blue area, A) or in the parenchyma (in the blue area, C) of the CFP-B16 tumors on different days after CTX-ACT treatment. Scale bar: 100 µm. (B,D) Trajectories of the individual CTLs at the periphery or in the parenchyma were plotted following the alignment of their starting positions. (E) Random walking analysis of

*Figure 3 continued on next page*

*Figure 3 continued*

the adoptive CTLs. Mean displacement (µm) versus the square root of the time (min$^{1/2}$) of the CTLs at the periphery of (top) or in the parenchyma (bottom) on different days, (**Figure 3—source data 1–9**). (**F–H**) Scatter plots of (**F**) the mean velocity, (**G**) the confinement ratio, and (**H**) the arrest coefficient of the CTLs at the tumor periphery or in the tumor parenchyma on different days after CTX-ACT treatment. Each data point represents a single cell, and the red bars indicate mean values. $*p<0.05$, $**p<0.01$, $***p<0.001$; ns – not significant, (**Figure 3—source data 10**). The data from 4–6 mice in three independent experiments were pooled.

The following source data is available for figure 3:

**Source data 1.** Mean displacement (µm) versus the square root of the time (min$^{1/2}$) of the CTLs at the tumor periphery on Day 1.
**Source data 2.** Mean displacement (µm) versus the square root of the time (min$^{1/2}$) of the CTLs at the tumor periphery on Day 3.
**Source data 3.** Mean displacement (µm) versus the square root of the time (min$^{1/2}$) of the CTLs at the tumor periphery on Day 5.
**Source data 4.** Mean displacement (µm) versus the square root of the time (min$^{1/2}$) of the CTLs at the tumor periphery on Day 6.
**Source data 5.** Mean displacement (µm) versus the square root of the time (min$^{1/2}$) of the CTLs in the tumor parenchyma on Day 3.
**Source data 6.** Mean displacement (µm) versus the square root of the time (min$^{1/2}$) of the CTLs in the tumor parenchyma on Day 5.
**Source data 7.** Mean displacement (µm) versus the square root of the time (min$^{1/2}$) of the CTLs in the tumor parenchyma on Day 6.
**Source data 8.** Linear fitting results of MD (mean displacement) of adoptive CTLs at the tumor periphery on Day 1, Day 3, Day 5 and Day 6.
**Source data 9.** Linear fitting results of MD (mean displacement) of adoptive CTLs in the tumor parenchyma on Day 3, Day 5 and Day 6.
**Source data 10.** Scatter plots of the mean velocity, confinement ratio, and arrest coefficient of the adoptive CTLs at the tumor periphery or in the tumor parenchyma on different days.

## CTX-ACT treatment decreases total endogenous T cells but retains activated endogenous CTLs in the tumor microenvironment

Next, we studied the dynamic behavior of endogenous CTLs in the tumor area. Here, we used *Cxcr6$^{+/gfp}$* transgenic mice that carried the CXCR6 sequence on one allele but had the GFP gene replacing the *Cxcr6* coding region on the other allele, in which GFP cells with the CD8$^+$ marker represent endogenous CTLs (*Unutmaz et al., 2000*; *Ruocco et al., 2012*).

First, the ex vivo analysis of endogenous GFP cells in the CFP-B16 tumors of mice treated with PBS, ACT, CTX, and CTX-ACT on Day 5 (11 days after implantation of CFP-B16 tumor cells) was performed by flow cytometry. The data showed that the endogenous GFP cells in the tumors decreased after CTX-ACT treatment (*Figure 4—figure supplement 1A,B*). Importantly, most of the GFP cells in the tumors of the CTX-ACT-treated mice were CD8$^+$ CTLs with expression of the activation marker CD69 (more than 60%, *Figure 4—figure supplement 1C*). These results suggest that the CTX-ACT combined treatment decreased the endogenous T cells but selectively retained the activated endogenous CD8$^+$ CTLs in the tumors.

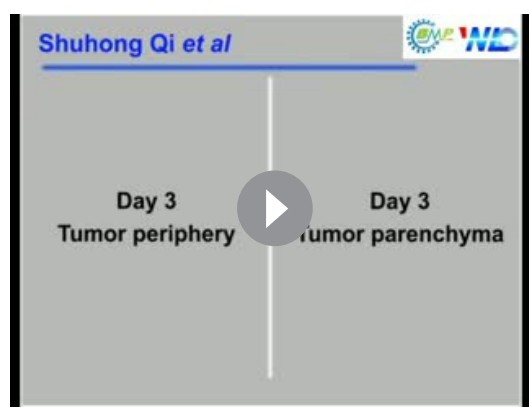

**Video 1.** In vivo sequential imaging of adoptive CTLs at the periphery and in the parenchyma of CFP-B16 tumors on Day 1 (after CTX-ACT treatment). The 3D time-lapse images were acquired as a 30 µm z-stack. Adoptively transferred CTLs are shown in green (CFSE-labeled), and the B16 tumor cells are shown in blue (CFP). Scale bar: 70 µm.

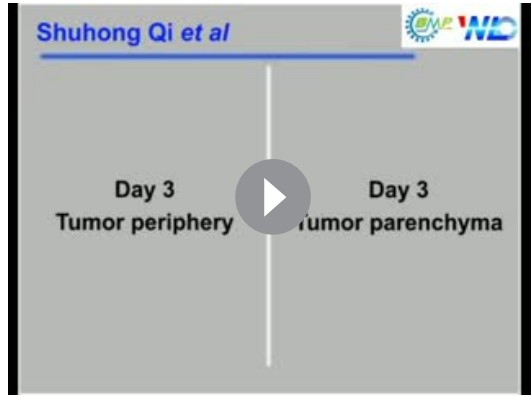

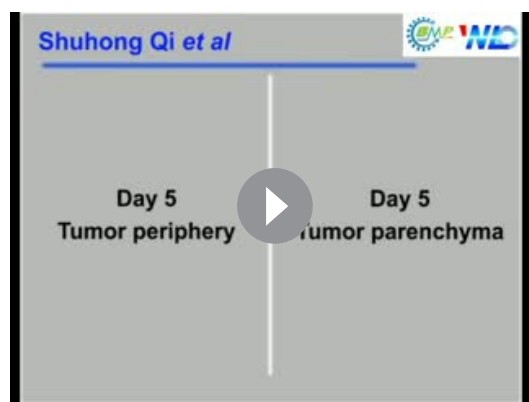

**Video 2.** In vivo sequential imaging of adoptive CTLs at the periphery and in the parenchyma of CFP-B16 tumors on Day 3. The 3D time-lapse images were acquired as a 30 μm z-stack. Adoptively transferred CTLs are shown in green (CFSE-labeled), and the B16 tumor cells are shown in blue (CFP). Scale bar: 70 μm. DOI: 10.7554/eLife.14756.027

**Video 3.** In vivo sequential imaging of adoptive CTLs at the periphery and in the parenchyma of CFP-B16 tumors on Day 5. The 3D time-lapse images were acquired as a 30 μm z-stack. Adoptively transferred CTLs are shown in green (CFSE-labeled), and the B16 tumor cells are shown in blue (CFP). Scale bar: 70 μm. DOI: 10.7554/eLife.14756.028

In order to distinguish adoptive and endogenous CTLs by intravital imaging, the adoptive CTLs were labeled with the red fluorescent dye CMTPX and then transferred into *Cxcr6*$^{+/gfp}$ transgenic mice with CFP-B16 tumor cells implanted into the window chamber on Day 0 (six days after tumor implantation). The intravital imaging was performed on Day 5 (five days after adoptive transfer of CTLs) because tumor elimination reached its peak in the CTX-ACT-treated mice on this day. Intravital imaging showed that the number of endogenous GFP T cells in the tumor area of CTX-ACT-treated mice was much smaller than that in ACT-treated mice (*Figure 4A,B*). The analysis of the migratory behavior of the endogenous T cells showed that, compared with the ACT group, the motility of the endogenous GFP T cells in the tumor areas of the CTX-ACT group was significantly decreased (*Video 5*), with more confined trajectories and a significantly increased arrest coefficient (mean velocity – 3.80 ± 2.56 μm min$^{-1}$ in the CTX-ACT group versus 4.6 ± 2.0 μm min$^{-1}$ in the ACT group; confinement ratio – 0.47 ± 0.26 in the CTX-ACT group versus 0.60 ± 0.25 in the ACT group; and arrest coefficient – 43 ± 35% in the CTX-ACT group versus 25 ± 26% in the ACT group; *Figure 4C–F*).

Both ex vivo characterization experiments and intravital imaging suggested that the CTX-ACT combined treatment deleted most of the endogenous T cells but retained the activated endogenous CTLs in the tumor area. The activated endogenous CTLs arrested in the tumor area for a long time and built stable, long-lasting interactions with the tumor cells to kill them efficiently.

## CTX-ACT treatment elicits the transient activation of endogenous TIls

Next, we used EGFP-transgenic C57BL/6 mice to observe the activated endogenous immunocytes in the tumor areas. CFP-B16 tumor cells were implanted into EGFP mice, in which all of the nucleated cells expressed EGFP and most of the mobile cells in vivo were immunocytes. Using

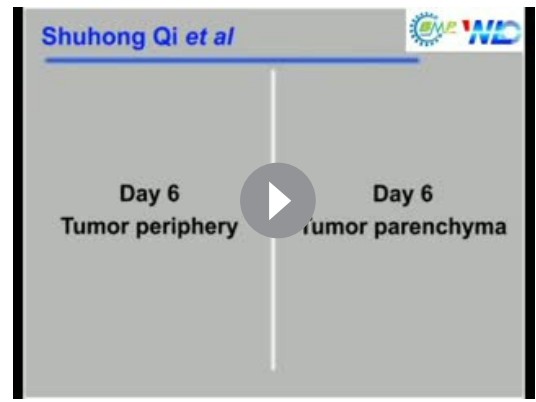

**Video 4.** In vivo sequential imaging of adoptive CTLs at the periphery and in the parenchyma of CFP-B16 tumors on Day 6. The 3D time-lapse images were acquired as a 30 μm z-stack. Adoptively transferred CTLs are shown in green (CFSE-labeled), and the B16 tumor cells are shown in blue (CFP). Scale bar: 70 μm. DOI: 10.7554/eLife.14756.029

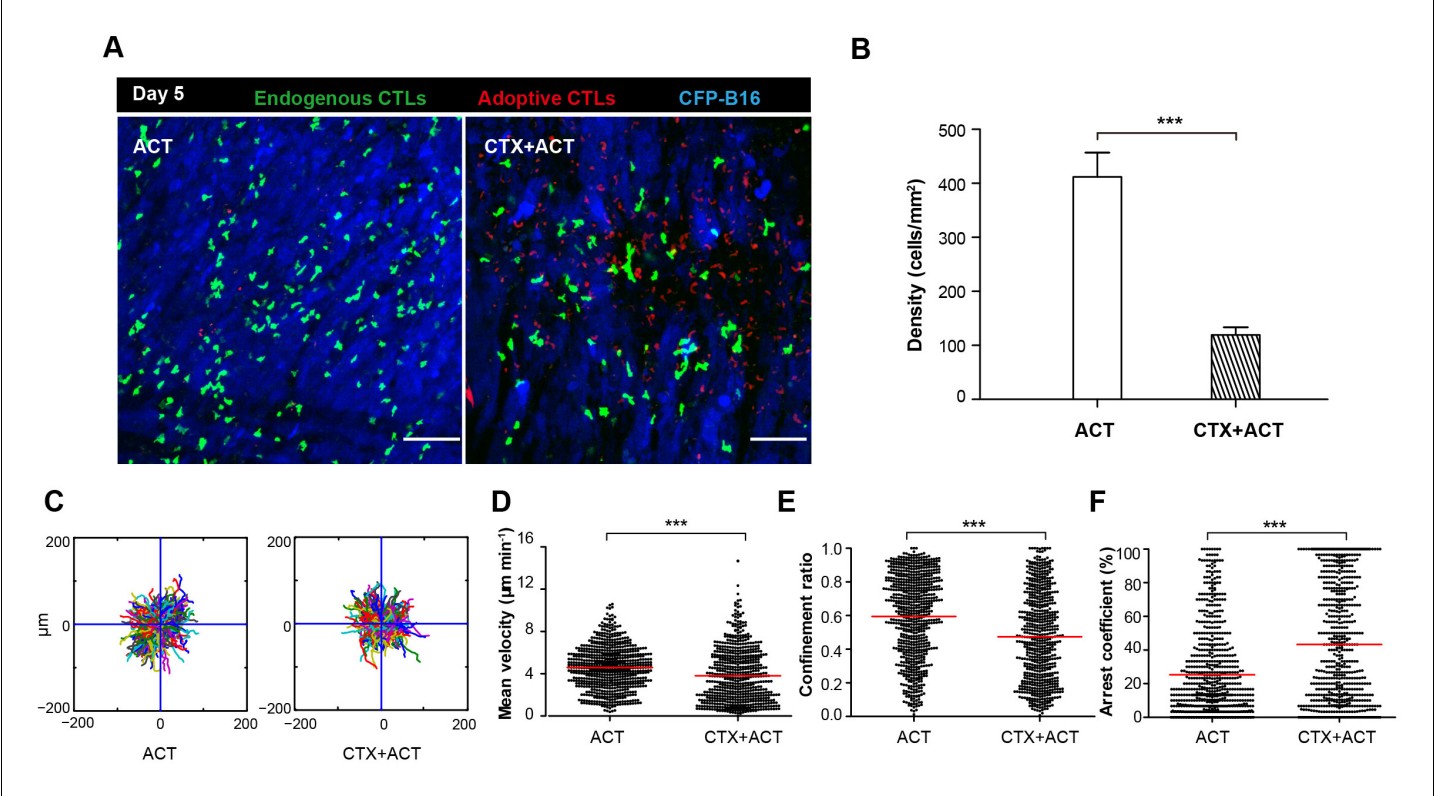

**Figure 4.** Migratory behavior of endogenous CTLs in the tumor microenvironment of mice treated with ACT and CTX-ACT on Day 5. (**A**) In vivo time-lapse images of the endogenous GFP T cells (green) and adoptive CTLs (red) in the CFP-B16 tumor area (blue). Mice were treated with ACT or CTX-ACT. Scale bar: 100 μm. (**B**) Quantification of endogenous GFP T cells in the differently treated groups on Day 5. Results are represented as the mean ± SEM (n = 11–19 fields, 0.40 mm² per field) from three mice per group. ***p<0.001 (**Figure 4—source data 1**). (**C**) Trajectories of GFP T cells in the differently treated groups were plotted following the alignment of their starting positions. (**D–F**) Scatter plots of (**D**) the mean velocity, (**E**) the confinement ratio, and (**F**) the arrest coefficient of the GFP T cells in tumor areas in the differently treated groups on Day 5. Each data point represents a single cell, and the red bars indicate mean values. *p<0.05, **p<0.01, ***p<0.001; ns: not significant (**Figure 4—source data 2**). The data from 3–5 mice in two independent experiments were pooled.

The following source data and figure supplement are available for figure 4:

**Source data 1.** Quantification of endogenous GFP T cells in the differently treated groups on Day 5.

**Source data 2.** Scatter plots of the mean velocity, confinement ratio, and arrest coefficient of the GFP T cells in tumor areas in the differently treated groups on Day 5.

**Figure supplement 1.** Characterization of GFP cells in differently treated *Cxcr6^{+/gfp}* mice.

a multi-photon excitation microscope, the tumor microenvironments of the CFP-B16 tumors and TIIs were continuously observed through a skin-fold window chamber from the day prior to the adoptive transfer of CTLs (Day 0–4, **Figure 5A,B**).

On Day 1, the endogenous TIIs in the CTX-ACT-treated mice migrated rapidly toward the tumor parenchyma (mean velocity – 4.92 ± 2.80 μm min$^{-1}$; n = 435 cells). The velocity on Day 1 (24 hr after CTX-ACT treatment, **Figure 5B–D**, **Video 6**) was approximately 2.4-fold faster than that on Day 0 (2.07 ± 2.18 μm min$^{-1}$, n = 477 cells; **Figure 5D**). Consistently, the TIIs on Day 1 displayed more expanded migration trajectories (**Figure 5C**) and less arrest (confinement ratio – 0.66 ± 0.25 on Day 1, 0.43 ± 0.26 on Day 0; arrest coefficient – 21 ± 21% on Day 1, 68 ± 37% on Day 0, **Figure 5E, F**).

On Day 2, the motility of the endogenous TIIs decelerated markedly (**Figure 5B–D**), demonstrating a decreased mean velocity (1.51 ± 1.78 μm min$^{-1}$; n = 136 cells) and confinement ratio (0.48 ± 0.29) and an increased arrest coefficient (75 ± 36%, **Figure 5D–F**). On Day 3 and Day 4, the

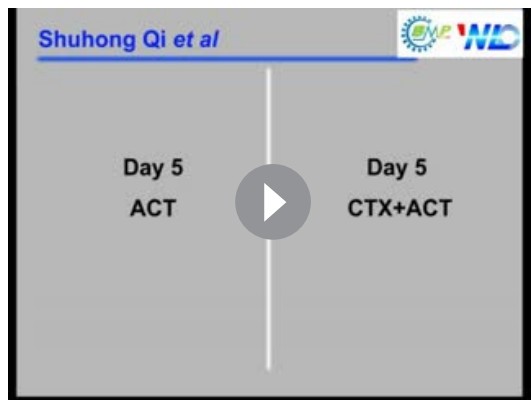

**Video 5.** In vivo sequential imaging of endogenous T cells and adoptive CTLs in CFP-B16 tumor areas in mice exposed to ACT and CTX-ACT combined treatments on Day 5. Time-lapse images were collected for 15 min. Endogenous T cells are shown in green (GFP labeled), adoptive CTLs are shown in red (CMTPX labeled) and the B16 tumor cells are shown in blue (CFP labeled). Scale bar: 100 μm.

speed of the TIIs further decreased (0.72 ± 0.31 μm min$^{-1}$, n = 164 cells; 0.62 ± 0.72 μm min$^{-1}$, n = 124 cells, respectively), and all other motility parameters of the TIIs differed from those on Day 2 (*Figure 5D–F*). The mean displacement of the TIIs on different days was consistent with random walking (*Figure 5G*). Thus, a transient increase in the motility of the endogenous TIIs was induced within 24 hr after the CTX-ACT treatment. These findings indicate that Day 1 is a key point in time for the activation of endogenous anti-tumor immune reactions in response to the CTX-ACT combined treatment. We speculated that the tumor destruction was caused by both adoptive CTLs and endogenous TIIs.

## Both CTX and ACT treatments are the essential factors for the chemotaxis and recruitment of TIIs into the tumor parenchyma

To further investigate the role of CTX-ACT treatment in the activation of endogenous TIIs, we analyzed the movement of TIIs in the tumor microenvironment of mice exposed to different treatments on the same day (Day 1 after the CTX-ACT treatment and seven days after tumor implantation). Analysis of the imaging data and migratory path indicated that the chemotaxis of the endogenous TIIs toward the tumor parenchyma was elicited only by the CTX-ACT treatment (*Figure 6A, B*, *Video 7*), which caused the TII movement to display long and linear trajectories (confinement ratio – 0.66 ± 0.25, *Figure 6B,D*). Compared with those in the CTX-ACT treatment group, the directions of TII movement in the ACT and PBS groups were disordered with confined trajectories (confinement ratio – 0.57 ± 0.27 in the ACT group and 0.47 ± 0.24 in the PBS group, *Figure 6A,B and D*). Although the trajectories of the TIIs in the CTX group were less confined with a high confinement ratio (0.63 ± 0.25), the directional movement of the TIIs was disordered (*Figure 6A,B*).

The endogenous TIIs in the CTX-ACT treatment group displayed the greatest motility with the fastest mean velocity (4.92 ± 2.80 μm min$^{-1}$; *Figure 6C*) and the lowest arrest coefficient (21 ± 21%, *Figure 6E*). The TIIs in both the ACT and PBS groups displayed similarly low mean velocities and high arrest coefficients (2.51 ± 2.53 μm min$^{-1}$ and 61 ± 40% in the ACT group, respectively; 2.18 ± 2.31 μm min$^{-1}$ and 66 ± 39% in the PBS group, respectively; *Figure 6C,E*). The motility of the TIIs in the CTX group was higher than that in the ACT or PBS groups but was still lower than that in the CTX-ACT group (mean velocity – 4.1 ± 3.1 μm min$^{-1}$; arrest coefficient – 39 ± 38%, *Figure 6C, E*). The mean displacement of TIIs in the different groups was consistent with random walking except for that in the CTX group, where the TIIs displayed slightly constrained motility (*Figure 6F*). Among all of these groups, only the CTX-ACT treatment group exhibited the movement of TIIs toward the tumor parenchyma with high motility and expanded trajectories. These findings indicate that the CTX-ACT treatment had a synergistic effect on the activation of endogenous TIIs, which in return accelerated the anti-tumor effects of the adoptive CTLs.

Immunofluorescence analysis of the frozen tumor sections showed that most of the tumor-surrounding EGFP cells were Ly6G$^+$ cells (*Figure 6—figure supplement 1A*). Hematoxylin and eosin (HE) staining further confirmed that more than 50% of the immunocytes surrounding the tumors were neutrophils (*Figure 6—figure supplement 1B and C*). These results indicated that the CTX-ACT treatment triggered the activity of endogenous neutrophils in the anti-tumor immune response rapidly, just 24 hr after the CTX-ACT treatment (Day 1).

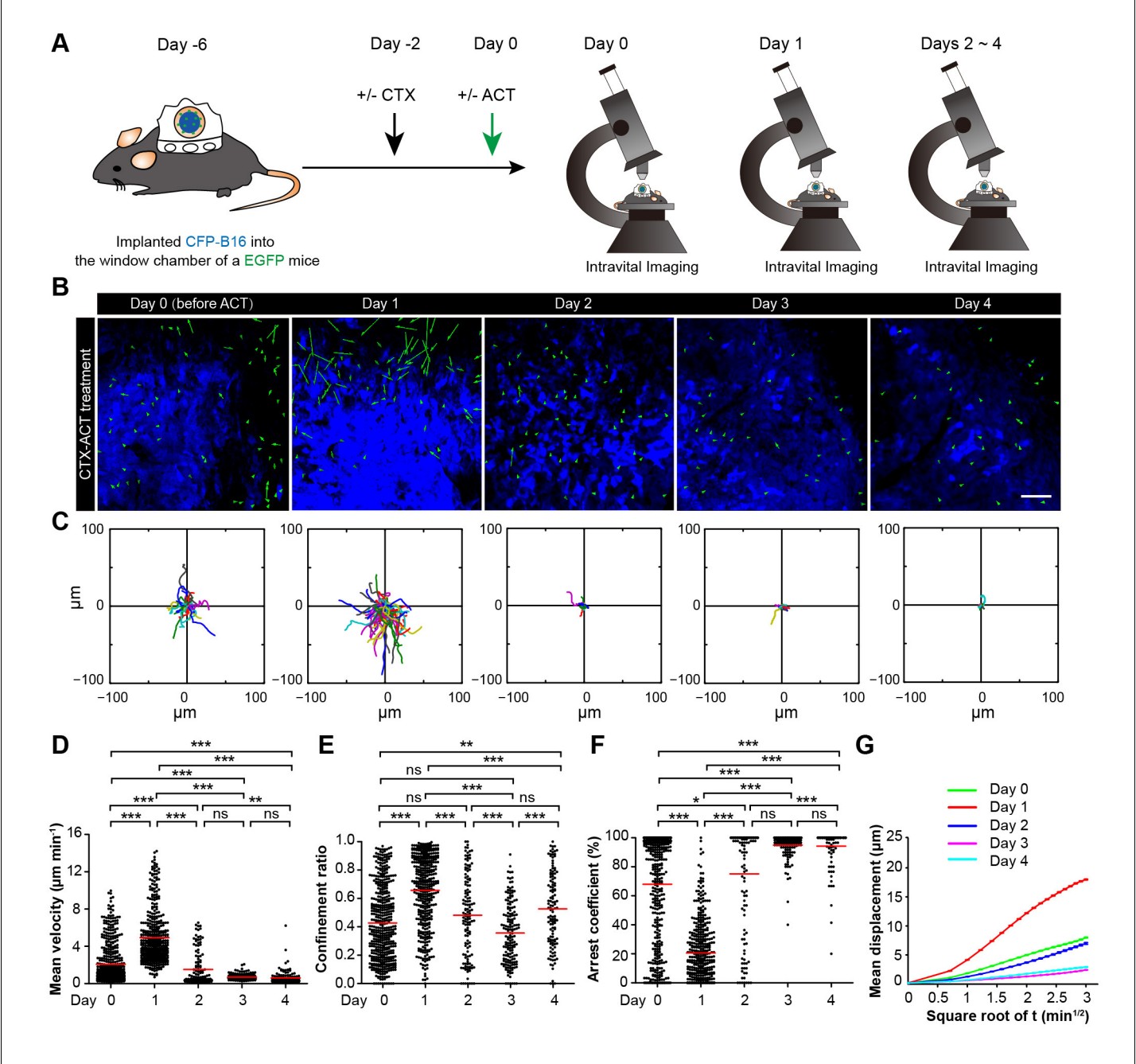

**Figure 5.** Migratory behavior of endogenous tumor-infiltrating immunocytes (TIIs) induced by the CTX-ACT treatment. (**A**) Experimental procedure for long-term intravital imaging of TIIs in the tumor microenvironment. (**B**) In vivo time-lapse images of EGFP TIIs (green) in the CFP-B16 tumor area (blue) from Day 0 to Day 4 after CTX-ACT treatment. Green arrows represent TIIs displacement, and blue areas represent CFP-B16 tumors. Scale bar: 100 µm. (**C**) Trajectories of individual TIIs on different days were plotted following the alignment of their starting positions. (**D–F**) Scatter plots of the (**D**) mean velocity, (**E**) confinement ratio, and (**F**) arrest coefficient of EGFP TIIs in tumor areas on different days. Each data point represents a single cell, and the red bars indicate mean values. *p<0.05, **p<0.01, ***p<0.001; ns, not significant, (**Figure 5—source data 1**). (**G**) Random walking analysis of the TIIs on different days. Mean displacement (µm) versus the square root of the time (min^1/2) of the TIIs, (**Figure 5—source data 2–7**). The data from 4–7 mice in three independent experiments were pooled.

The following source data is available for figure 5:

**Source data 1.** Scatter plots of the mean velocity, confinement ratio, and arrest coefficient of EGFP TIIs in tumor areas on different days.

*Figure 5 continued on next page*

*Figure 5 continued*

**Source data 2.** Mean displacement (μm) versus the square root of the time ($min^{1/2}$) of the TIIs on Day 0.
**Source data 3.** Mean displacement (μm) versus the square root of the time ($min^{1/2}$) of the TIIs on Day 1.
**Source data 4.** Mean displacement (μm) versus the square root of the time ($min^{1/2}$) of the TIIs on Day 2.
**Source data 5.** Mean displacement (μm) versus the square root of the time ($min^{1/2}$) of the TIIs on Day 3.
**Source data 6.** Mean displacement (μm) versus the square root of the time ($min^{1/2}$) of the TIIs on Day 4.
**Source data 7.** Linear fitting results of MD (Mean displacement) of TIIs at tumor areas on Day 0–Day 4.

## CTX-ACT treatment accelerates the infiltration of mature DCs into the tumor areas

DCs also played a key role in activating anti-tumor immune responses. To investigate whether the CTX-ACT treatment accelerated the infiltration of DCs into the tumors, we observed the tumor microenvironment using large-field microscopy. The imaging data showed that many DCs infiltrated into the tumor areas of the CTX-ACT-treated mice on Day 3 (three days after CTX-ACT treatment, *Figure 7A*). The density of the DCs in the CTX-ACT group increased by 3.6-fold compared with that in the PBS group, by 2.0-fold compared with that in the CTL group and by 1.6-fold compared with that in the CTX group (*Figure 7B*). Immunofluorescence analysis of the frozen tumor sections from CTX-ACT-treated mice revealed that certain YFP-labeled DCs displayed an mature phenotype characterized by the expression of MHC-II and CD86, indicating that some mature DCs were present in the tumor areas (*Figure 7C*). This finding suggests that the CTX-ACT treatment promotes the infiltration of DCs (some of them were mature) to the tumor areas to further enhance the anti-tumor effects.

## Design of the metronomic schedule of the CTX-ACT treatment according to the information acquired by intravital imaging

Although the CTX-ACT treatment only controlled the tumor growth for several days and displayed no significant difference compared with the CTX treatment group (*Figure 2A*), intravital imaging revealed that the CTX-ACT combined treatment elicited a stronger anti-tumor immune response than the CTX treatment alone. To improve the anti-tumor efficacy of the combined immunotherapy, we adopted three rounds of CTX-ACT treatment using a metronomic schedule (*Figure 8A*). According to the intravital imaging data, the motility of the adoptive CTLs remained at a high level on Day 5 both at the tumor periphery and in the parenchyma (*Figure 3A–D*, *Video 3*). Concomitant with this phenomenon, the anti-tumor response also peaked on Day 5 (*Figure 2C*, *Figure 3A,C*), and then the tumor continued to grow (*Figure 2C*). Before the adoptive CTLs were recruited and activated, the motility of the endogenous TIIs transiently increased on Day 1 and then the TIIs efficiently infiltrated into the tumor parenchyma (*Figure 5B*, *Video 6*). According to these results, an effective interval time of five days between treatment rounds was determined for the CTX-

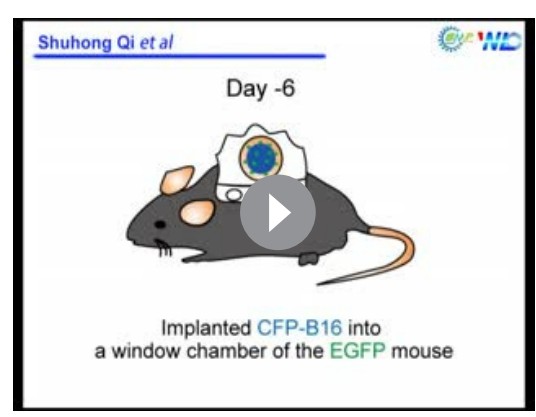

**Video 6.** In vivo sequential imaging of endogenous TIIs at the periphery of the CFP-B16 tumors on different days following treatment with CTX and ACT. Time-lapse images were collected for 10 or 15 min. Endogenous TIIs are shown in green (EGFP), and the B16 tumor cells are shown in blue (CFP). Scale bar: 70 μm.

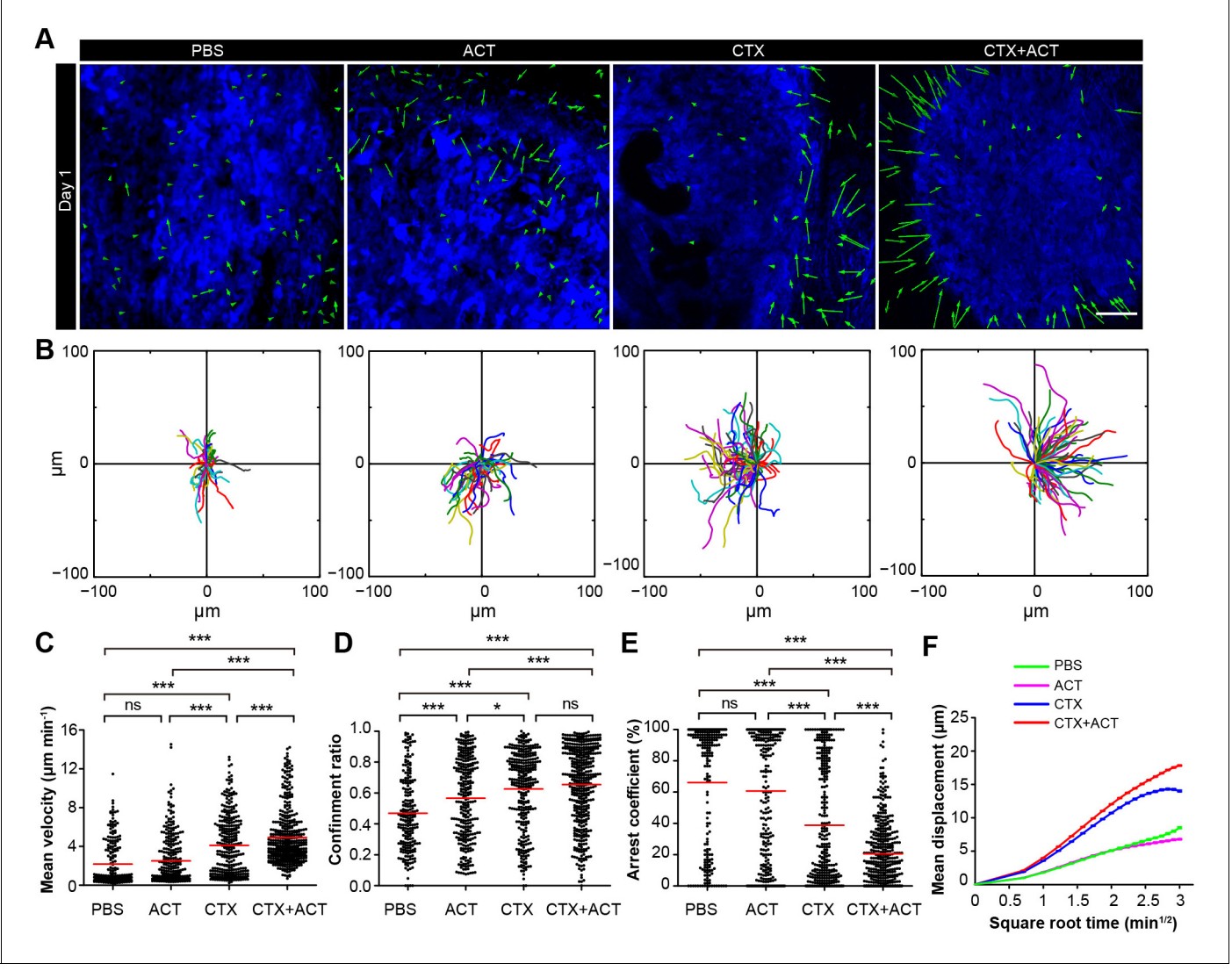

**Figure 6.** Migratory behavior of the TIIs following different treatments. (A) In vivo time-lapse images of the EGFP TIIs in the CFP-B16 tumor area on Day 1. Mice were treated with ACT, CTX, CTX-ACT or PBS control. Green arrows represent TIIs displacement, and blue areas represent CFP-B16 tumors. Scale bar: 100 μm. (B) The trajectories of individual EGFP TIIs in different treated groups were plotted following the alignment of their starting positions. (C–E) Scatter plots of the (C) mean velocity, (D) confinement ratio, and (E) arrest coefficient of the EGFP TIIs in the differently treated groups. Each data point represents a single cell, and the red bars indicate mean values. *p<0.05, **p<0.01, ***p<0.001; ns, not significant (*Figure 6—source data 1*). (F) Random walking analysis of the TIIs in the different groups. Mean displacement (μm) versus the square root of time (min$^{1/2}$) of TIIs in different treatment groups ( *Figure 6—source data 2–6*). The data from 12–15 mice in three independent experiments were pooled.

The following source data and figure supplements are available for figure 6:

**Source data 1.** Scatter plots of the mean velocity, confinement ratio, and arrest coefficient of the EGFP TIIs in the different treatment groups.
**Source data 2.** Mean displacement (μm) versus the square root of the time (min$^{1/2}$) of the TIIs in the PBS group.
**Source data 3.** Mean displacement (μm) versus the square root of the time (min$^{1/2}$) of the TIIs in the ACT group.
**Source data 4.** Mean displacement (μm) versus the square root of the time (min$^{1/2}$) of the TIIs in the CTX group.
**Source data 5.** Mean displacement (μm) versus the square root of the time (min$^{1/2}$) of the TIIs in the CTX-ACT group.
**Source data 6.** Linear fitting results of MD (Mean displacement) of TIIs at tumor areas in the different treatment groups on Day 1.

*Figure 6 continued on next page*

*Figure 6 continued*

**Figure supplement 1.** Phenotype of EGFP TIIs in CFP-B16 tumors of mice following different treatments.
**Figure supplement 1—source data 1.** Percentage of neutrophils among TIIs at the periphery of the tumors in mice that received different treatments.

ACT combined therapy. When the anti-tumor effect of the adoptive CTLs decreased, a second round of treatment was implemented to elicit the transient activity of the endogenous TIIs. This approach maintained a specific immune reaction against CFP-B16 tumors in vivo at high levels. When administered three times successfully, the metronomic treatments controlled the growth of the tumor during its entire course, with a significantly more effective tumor growth control compared with the other control groups that received only a single round of the CTX-ACT treatment (p<0.001, *Figure 8B*) and the metronomic CTX treatment (p<0.05, *Figure 8B*).

The entire anti-tumor immune response elicited by the three rounds of metronomic CTX-ACT treatments was captured by long-term intravital imaging. As shown in *Figure 8C*, a conflict occurred between the tumor cells and immunocytes. Tumor growth was controlled for four days after the first round of CTX-ACT treatment (*Figure 8C*, top row). When the tumor began to regrow one day later, the second treatment was applied, and continued to control the growth of the tumor for an additional four days (*Figure 8C*, middle row). The third treatment was applied when the tumor began to regrow, and controlled the growth of the tumor successfully again for several more days (*Figure 8C*, bottom row). During this process, the continuous elimination of Tregs, transfusion of adoptive CTLs and activation of endogenous immunocytes (*e.g.*, CTLs, neutrophils and DCs) were required to control the tumor growth.

## Discussion

To understand the cellular mechanisms underlying tumor immunotherapy, we globally monitored the sequence of events involving multicolor-coded immune cells during CTX-ACT immunotherapy using a tumor-implanted window chamber device (*Schietinger et al., 2013*) and large-field intravital imaging technology. To the best of our knowledge, this study is the first to elucidate the entire process and to provide sequential descriptions of the dynamic spatio-temporal changes in Tregs, adoptive CTLs, endogenous CTLs, endogenous TIIs and DCs in the tumor microenvironment (*Figure 9*). In particular, it is worth mentioning that we discovered two symbolic cell events that characterize immunosuppression and immunoactivation: the formation of an immunosuppressive ring of Tregs around the solid tumor, which indicates immune suppression, and the chemotactic movement of endogenous TIIs and infiltration of adoptive CTLs and DCs into the tumor parenchyma, which demonstrate immune activation.

An immunosuppressive environment was caused by the accumulation of a large number of Tregs in the tumor areas. Tregs are considered to play a crucial role in mediating immunosuppression in anti-tumor immune responses (*Zou, 2006*; *Beyer and Schultze, 2006*). By using large-field intravital imaging, we observed that Tregs not only concentrated at the tumor periphery (*Figure 1—figure supplement 3*) but also formed an immunosuppressive ring around a solid tumor (*Figure 1C,D*), and adoptive CTLs barely infiltrated solid tumors in the presence of this immunosuppressive ring (*Figure 1C, D*,

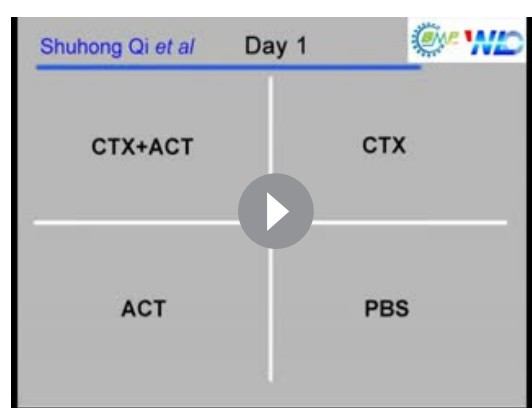

**Video 7.** In vivo sequential imaging of endogenous TIIs at the periphery of the CFP-B16 tumors on Day 1 in mice exposed to different treatments. Time-lapse images were collected for a duration of 10 or 15 min. Mice were mock-treated (with PBS) or treated with CTX, ACT, or CTX-ACT as indicated. Endogenous TIIs are shown in green (EGFP-labeled) and the B16 tumor cells are shown in blue (CFP-labeled). Scale bar: 70 μm.

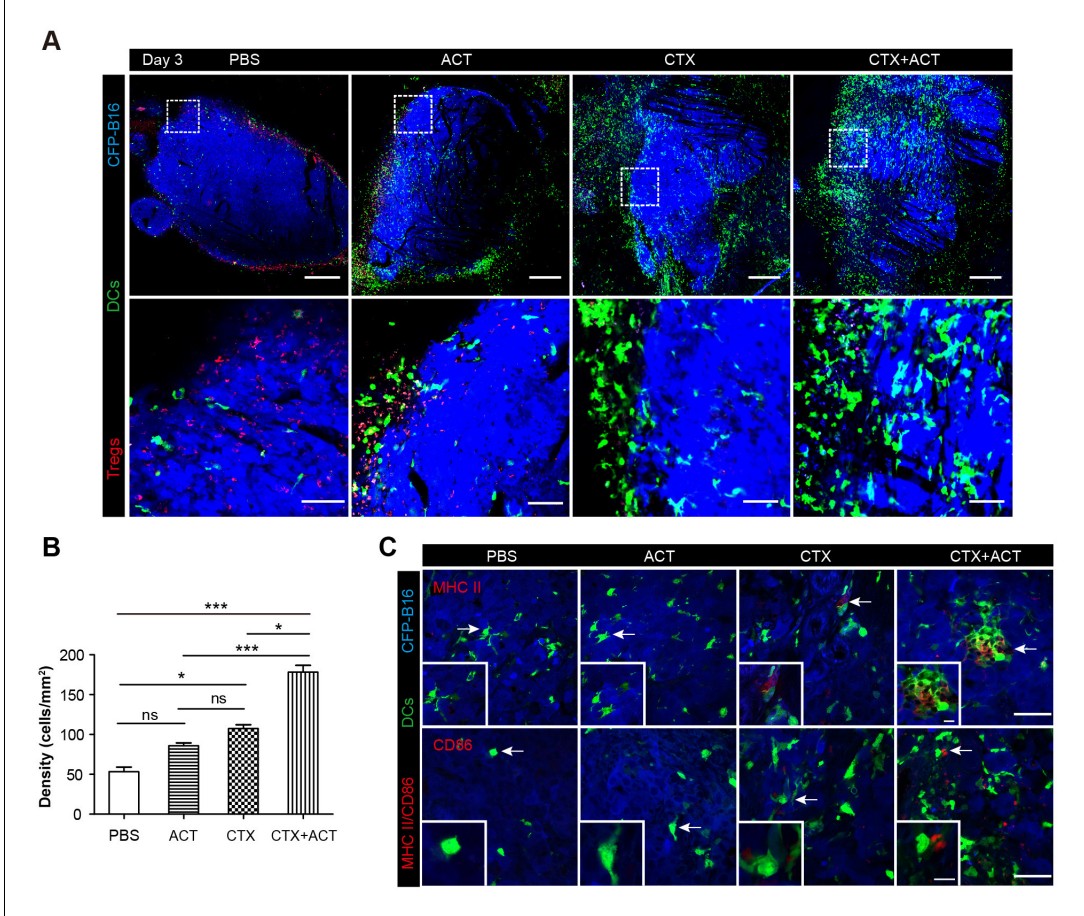

**Figure 7.** Intravital imaging of DCs infiltrating into the tumor areas of the mice following different treatments. (A) Large-field intravital imaging of DCs (green) and Tregs (red) in the CFP-B16 tumor area (blue) on Day 3. Top row: large-field images; scale bar 500 μm. Bottom row: images of the region of interest from the top row; scale bar 100 μm. (B) Density of DCs in the tumor areas in the different treatment groups. The data are represented as the mean ± SEM (n = 10–14 fields, 12 mm² per field (large-field images) or 0.40 mm² per field) from three independent experiments. *p<0.05, **p<0.01, ***p<0.001; ns, not significant (*Figure 7—source data 1*). (C) Representative images of mature DCs in tumor sections were immunofluorescently labeled to detect MHC II (top row) and CD86 (bottom row). Scale bar: 40 μm. Inserts are magnifications of the regions indicated with arrows. Scale bar: 15 μm.

The following source data is available for figure 7:

**Source data 1.** Density of DCs in the tumor areas in the different treatment groups.

*Figure 2B*, and *Figure 1—figure supplement 3*). CTLs must recognize the cognate antigen of tumor cells before they are able to infiltrate into the tumor parenchyma sufficiently (*Boissonnas et al., 2007*; *Mrass et al., 2006*). Thus, we propose that the immunosuppressive ring of Tregs blocked the ability of adoptive CTLs to recognize the cognate antigen of tumor cells and decreased the migration of these CTLs into the tumor area. This conclusion is supported by the increased number of adoptive CTLs infiltrating into the tumor parenchyma after the CTX treatment (*Figure 2B,C*), which depleted most of the Tregs in the tumor area and blocked the formation of an immunosuppressive ring around the solid tumor (*Figure 2B,C*). Additionally, the depletion of the Tregs induced by CTX treatment contributed to the accumulation of adoptive CTLs in the tumor area in three ways: (1) it cleared the immunosuppressive barrier to facilitate the migration of adoptive CTLs into the tumor area; (2) it induced the elevated expression of some chemokines and cytokines in the tumor microenvironment to promote the accumulation of adoptive CTLs (*Bracci et al., 2007*; *Schiavoni et al., 2011*); and (3) it provided space to allow the infiltration and homeostatic proliferation of adoptive CTLs (*Bracci et al., 2007*; *Sistigu et al., 2011*).

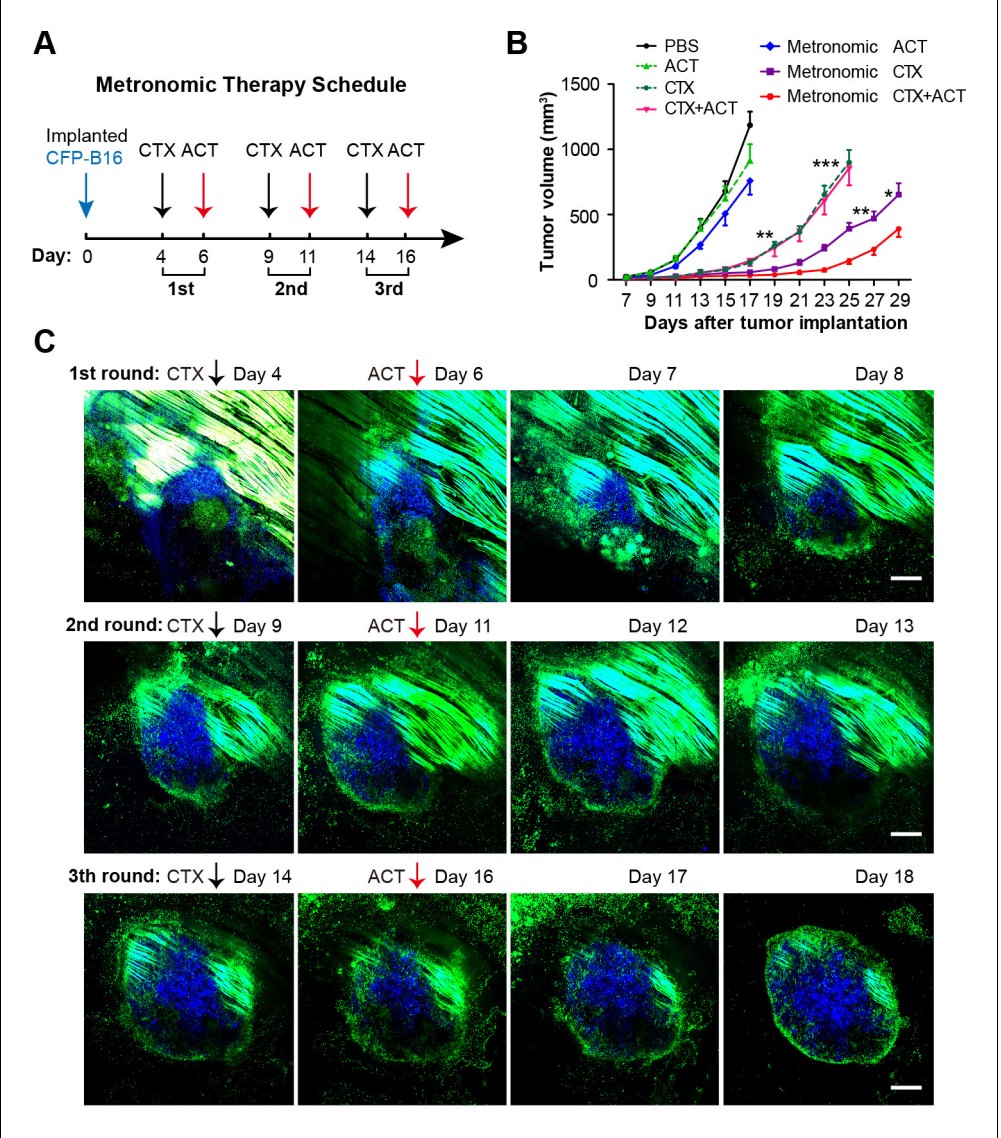

**Figure 8.** Metronomic CTX-ACT therapy efficiently controlled the growth of CFP-B16 tumors in vivo. **(A)** Metronomic therapy schedule of the CTX-ACT treatment. **(B)** Growth curves for CFP-B16 tumors in the different treatment groups. The data are represented as the mean ± SEM tumor volume (n = 12–15, three independent experiments). *p<0.05, **p<0.01, ***p<0.001(*Figure 8—source data 1*). **(C)** Long-term and large-field intravital images of the tumor microenvironment during CTX-ACT metronomic therapy. Blue, a CFP-B16 tumor; green, EGFP host cells. Scale bar: 500 μm.

The following source data is available for figure 8:

**Source data 1.** Growth curves for CFP-B16 tumors in the different treatment groups

As shown in the schematic in *Figure 9*, specific endogenous and adoptive anti-tumor immune reactions were triggered by the synergistic effect of the CTX-ACT combined treatment. Endogenous TILs were activated and underwent chemotactic movements toward the tumor parenchyma within 24 hr (*Figure 5B*, *Video 6*), before endogenous DCs infiltrated into the tumor parenchyma on Day 3 (*Figure 7A,C*). In addition, adoptive CTLs accumulated at the tumor periphery, arrested in the tumor parenchyma to contact neighboring tumor cells on Day 3 and then resumed their high-speed migration on Day 5 (*Figure 3A,C and F*). On Day 5, the anti-tumor effect elicited synergistically by CTX and ACT in a combined treatment achieved its maximum and promoted external and internal

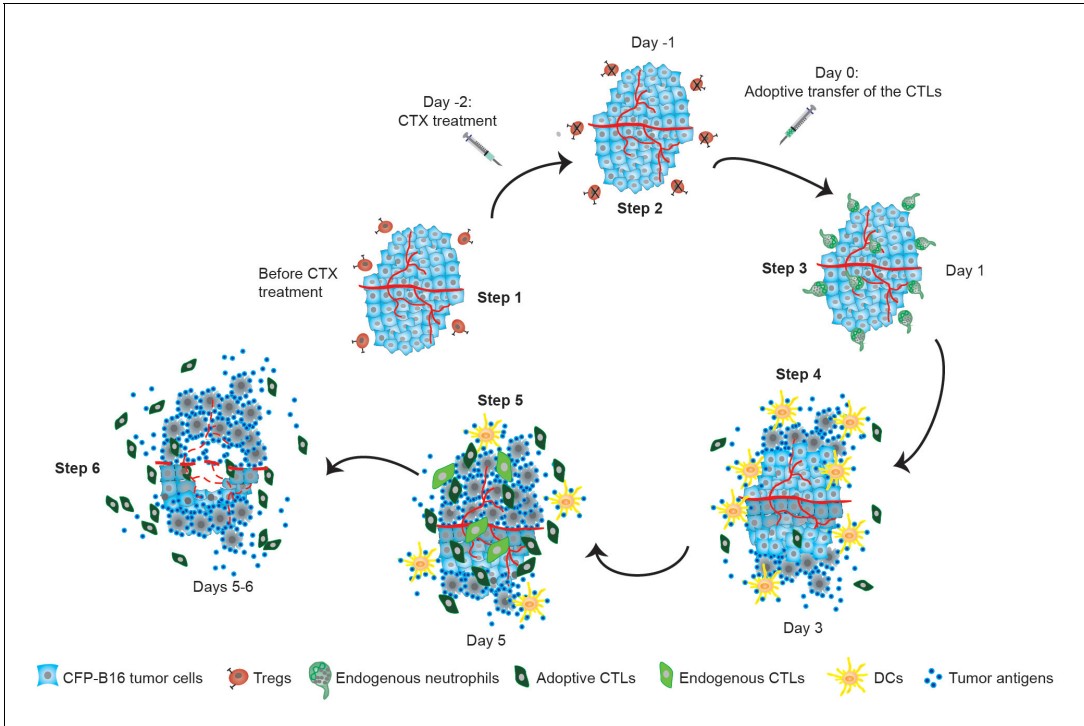

**Figure 9.** Timeline schematic showing the elicitation of anti-tumor immune responses in the tumor microenvironment by CTX-ACT treatment. Step 1: CFP-B16 tumor cells grew and Tregs accumulated in the tumor area before CTX treatment. Step 2: CTX treatment depleted most Tregs, blocking the formation of an 'immunosuppressive ring'. Step 3: A transient increase in the motility of the endogenous neutrophils is elicited by CTX-ACT treatment on Day 1. Step 4: DCs present increased infiltration on Day 3. Step 5: Adoptive CTLs present increased infiltration and motility and the CTX-ACT treatment retained activated endogenous CTLs in the tumor area on Day 5. Step 6: Solid tumor 'melting' occurs from the inside and 'shrinking' occurs from the outside on Days 5–6.

tumor shrinking and melting (*Figure 2C*, *Figure 3A,C*). Our results highlight the importance of metronomic treatments in CTX-ACT therapy, as a designed metronomic CTX-ACT treatment schedule (*Figure 8A*) alternately accelerates the endogenous and adoptive anti-tumor immune responses and successfully controls the tumor growth for a long duration (*Figure 8B,C*).

The migratory behavior of CTLs in the tumor microenvironment is complicated and changes dynamically. Certain researchers have indicated that T cells need to arrest for a long period at the tumor periphery and that their interaction with tumor cells is crucial for tumor elimination (*Boissonnas et al., 2007*). However, other studies have shown that the random, high speed migration of lymphocytes from the tumor periphery to the tumor center is beneficial for tumor elimination (*Mrass et al., 2006*). In our study, we observed four different stages of adoptive CTL migration in the same tumor region, which occurred during the tumor destruction process over a period of up to seven days after the CTX-ACT combined treatment (*Figure 3A, C*, and *Videos 1–4*). The adoptive CTLs accumulated at the tumor periphery but barely infiltrated into the tumor parenchyma during the first stage (*Figure 3A-D*, *Video 1*). Then, the CTLs formed a long-term interaction with the tumor cells in both the tumor periphery and the tumor parenchyma during the second stage (*Figure 3A-D*, *Video 2*). Simultaneously with tumor cell death and elimination, the adoptive CTLs resumed their motility during the third stage (*Figure 3A-D*, *Video 3*). Finally, the number and motility of the CTLs rapidly decreased, suggesting CTL dysfunction during the fourth stage (*Figure 3A-D*, *Video 4*). Thus, the differences in the behavior of the adoptive CTLs throughout the four stages provide an explanation for the variations observed in the previous reports. Besides the activities of the adoptive CTLs, the dynamic behavior of endogenous CTLs has also been observed in our study. The CTX-ACT combined treatment depleted most of the endogenous T cells in the tumor area, and most of the remaining endogenous T cells were activated CD8[+] CTLs (*Figure 4—figure supplement 1A–C*). While the adoptive CTLs had complex migratory behaviors during the four different stages, the

endogenous CTLs arrested at the tumor area for a long time and showed confined movement so that they formed stable interactions with tumor cells and then effectively destroyed them (*Figure 4A,C–F*).

In recent years, the interest in tumor-associated neutrophils (TANs) has increased, and accumulating data have demonstrated that TANs play dichotomous roles because they exhibit both pro-tumor and anti-tumor effects during tumor progression (*Mantovani et al., 2011*; *Piccard et al., 2012*; *Brandau et al., 2013*). Previous studies have demonstrated that during the early stage of tumor progression, accumulating TANs with high motility ((*Yao et al., 2015*), also described as TENs (*Granot et al., 2011*)), play an important role in the anti-tumor response by exerting a direct cytotoxic effect on tumor cells (*Hicks et al., 2006*), enhancing antibody-induced anti-tumor effects (*Albanesi et al., 2013*), and having a stimulatory effect on T cell proliferation (*Eruslanov et al., 2014*). In the present study, more than 50% of the TIIs were neutrophils at a very early stage (Day 1) after the CTX-ACT treatment (*Figure 6—figure supplement 1*) and exhibited a high motility. Thus, we assumed that these neutrophils were TENs with anti-tumor effects. After the CTX-ACT combined treatment, the endogenous TENs transiently moved (within 24 hr) with high speed via chemotaxis toward the tumor parenchyma (*Figure 5*, *6*), and their motility increased before the accumulation of adoptive CTLs and DCs (*Figure 2C*, *7A*). Intravital imaging showed that the number of DCs increased significantly on Day 3 after the CTX-ACT treatment (*Figure 7A,B*). Some DCs displayed mature phenotypes in the tumor tissues of CTX-ACT-treated mice and presented high MHC-II and CD86 expression (*Figure 7C*). A large number of DCs infiltrating into the tumor parenchyma has been shown to have a positive association with longer survival of cancer patients (*Chaput et al., 2008*; *Liu et al., 2005*; *Pagès et al., 2010*). Notably, the recruitment of DCs into the tumor area occurred earlier than the solid tumor 'melting' from the inside. This phenomenon suggests that the infiltrated DCs may take up tumor antigens and present them to T cells (*Restifo et al., 2012*) to further elicit the anti-tumor immune response.

Most previous studies applied the OT-I system with $K^b$-OVA as a model antigen in T lymphoma cells to observe the migratory behavior of antigen-specific CTLs in solid tumors (*Boissonnas et al., 2007*; *Breart et al., 2008*; *Boissonnas et al., 2013*). However, cancer cell antigens are expressed at a low level in most patients (*Restifo et al., 2012*). Thus, the $K^b$-OVA antigen tumor model could not completely mimic the clinical presentation. In the present study, by contrast, we designed an experimental system to mimic ACT immunotherapy using whole tumor cell antigens to prime and activate the CTLs.

In summary, the CTX-ACT combined treatment displayed synergistic anti-tumor effects by blocking the formation of an immunosuppressive ring around solid tumors by Tregs, eliciting the transient migratory activity of TIIs toward the tumor parenchyma, and accelerating the infiltration of adoptive CTLs and endogenous DCs into the tumor. The CTX-ACT treatment not only promoted the anti-tumor effects of adoptive CTLs but also elicited endogenous anti-tumor immune responses. Guided by the results obtained by intravital imaging, three rounds of metronomic CTX-ACT treatments successfully controlled the tumor growth for several weeks. Thus, our long-term intravital imaging of the multicolor-coded tumor microenvironment provides new perspectives regarding the cellular mechanisms underlying the success or failure of cancer immunotherapies,and will facilitate improvements in the efficacy of combination immunotherapy. Furthermore, the method of long-term intravital imaging of various immune cells in the tumor microenvironment in vivo is also suitable for monitoring multiple immune reactions, studying the spatio-temporal cellular events of other immunotherapies, such as checkpoint inhibitors (CTLA-4 antibody and PD-1/PD-L1 antibodies), and comparing the efficacy of classical immunotherapies (such as the CTX-ACT combined treatment) and checkpoint inhibitors in the treatment of different tumor models.

## Materials and methods

### Mice

C57BL/6 female mice and BALB/c nude mice (of 6–12 weeks old) were obtained from Hunan Slack King of Laboratory Animal Co., Ltd (Hunan, China). B6.Cg-*Tg (Itgax-Venus) 1Mnz*/J (*CD11c*-YFP, RRID: IMSR_JAX:008829) mice, C57BL/6-*Foxp3*$^{tm1Flv}$/J (*Foxp3*-mRFP, RRID: IMSR_JAX:008374) mice and B6.129P2-*Cxcr6*$^{tm1Litt}$/J (*Cxcr6*$^{+/gfp}$, RRID: IMSR_JAX:005693) mice were derived from breeding

pairs that were originally obtained from Jackson Laboratory (Bar Harbor, ME). To generate multi-color-coded transgenic mice, *CD11c*-YFP mice were hybridized with *Foxp3*-mRFP mice. C57BL/6-*Tg (CAG-EGFP)*/J mice, that express EGFP throughout the entire body, excluding erythrocytes and hair, were generously provided by Dr. Zhiying He (Second Military Medical University, Shanghai, China). All of the mice were bred and maintained in a specific pathogen-free (SPF) barrier facility at Animal Center of Wuhan National Laboratory for Optoelectronics. All animal studies were approved by the Hubei Provincial Animal Care and Use Committee and followed the experimental guidelines of the Animal Experimentation Ethics Committee of Huazhong University of Science and Technology.

## Tumor cells

B16 melanoma cells were purchased from Boshide Biology Ltd. China (RRID: CVCL_F936). The B16 cell line was stably transfected with the PB transposon system (*Ding et al., 2005*) (a gift from Dr. Xiaohui Wu, Fudan University, Shanghai, China), which contained the sequence encoding mCerulean to generate the CFP-B16 tumor cell line. All cell lines were mycoplasma-negative as determined by screening using the MycoProbe Mycoplasma Detection Kit (R and D Systems, Minneapolis, MN). The CFP-B16 cell line was authenticated using the Cell Line Authentication Service by short-tandem repeat (STR) profiling carried out by Beijing Microread Genetics Co., Ltd. (Beijing, China). These cells were cultured in RPMI-1640 medium (HyClone, Beijing, China) containing 1% penicillin-streptomycin (HyClone) and 10% fetal bovine serum (FBS, HyClone).

## Production of CFP-B16 reactive T cells

T cells that were reactive with CFP-B16 were established from C57BL/6 mice using an immunization strategy consisting of a series of tumor cellchallenges. First, the C57BL/6 mice were immunized sub-cutaneously in both flanks with $2.5 \times 10^6$ CFP-B16 cells (pretreated with 50 µg ml$^{-1}$ mitomycin C [Sigma-Aldrich, Saint Louis, MO] for 2 hr at 37°C). Seven days after the primary immunization, the mice were immunized using the same methods. Seven days after the rechallenge, the mice were euthanized, and their spleens were dissected to prepare immunocytes. Spleen-derived cells (1–$2 \times 10^6$ per ml) were cultured in 24-well plates (Costar, Suzhou, China) with RPMI-1640 medium (HyClone), 10% FBS (HyClone), IL-2 (50 U ml$^{-1}$, Peprotech, Rocky Hill, NJ, USA) and CFP-B16 whole-cell antigen (50 µg ml$^{-1}$, supernatant from freeze-thawed tumor cell lysate). Three days later, when the CTLs became confluent, the cells were split 1:2 to 1:4 into new 24-well plates using fresh complete medium with whole-cell antigen and IL-2. The method of generating CFP-B16 CTLs in vitro is discussed in the book 'Current protocols in Immunology' (*Restifo and Nicholas, 2011*) and in previous reports (*Liu et al., 2006*; *Bauer et al., 2014*).

## Cytotoxicity assays and flow cytometry

The cytotoxicity of the CTLs was evaluated using CFSE and PI dual-staining assays. Target cells (CFP-B16 and normal C57BL/6 splenocytes) were labeled with 5 µM CFSE (Invitrogen, Eugene, OR) for 15 min at 37°C. The CTLs were collected from cultured lymphocytes by centrifugation in Histopaque−1.083 (Sigma-Aldrich, Saint Louis, MO) and co-cultivated with CFSE-labeled target cells in 96-well U-bottom plates (Costar, Suzhou, China) at different E:T ratios (25:1, 12.5:1, 5:1, 2.5:1, 1.25:1, 0.5:1, and 0.25:1). After the cells were incubated for 4 hr at 37°C, the mixed cells were labeled with 5 µM PI (Sigma-Aldrich, Saint Louis, MO) in the dark for 10 min. The dual fluorescent signals of the target cells were analyzed using a FACS-Calibur flow cytometer (Guava EasyCyte 8HT, EMD Millipore Corporation, Darmstadt, Germany).

The percentage of CTL-specific lysis was calculated according to the following formula:

Cytotoxicity (%) = [(experimental dead target cells (%) − spontaneous dead target cells (%)) / (1 − spontaneous dead target cells (%))]×100.

Before the adoptive transfer, the activity and quality of a number of CTLs cultured in vitro were tested. The CTLs were incubated with anti-mouse CD 16/32 (Fc block, Cat# 101302, Biolegend, San Diego, CA; RRID: AB_312801) for 10 min, and then they were stained with various antibodies at 4°C in the dark for 30 min and washed once with PBS (HyClone). The supernatants were discarded, and the cell pellets were resuspended in 0.2 ml PBS before analysis using a FACS-Calibur flow cytometer (Guava EasyCyte 8HT). Typically, 10,000 events were assessed. For the intracellular cytokine staining, the CTLs were stimulated with whole CFP-B16 cell antigen in the presence of Brefeldin A (10 µg

ml$^{-1}$, eBioscience, San Diego, CA), and the cells were maintained in Brefeldin A until fixation and then prepared as previously described. CD3-PE (Cat# 100308, RRID: AB_312673), CD3-APC/Cy7 (Cat# 100329, RRID: AB_1877171), CD8-FITC (Cat# 100705, RRID: AB_312744), CD8-PE/Cy7 (Cat# 100722, RRID: AB_312761), CD4-APC (Cat# 100412, RRID: AB_312697), CD25-FITC (Cat# 102005, RRID: AB_312854), CD69-PerCP/Cy5.5 (Cat# 104521, RRID: AB_940497), CD11b-APC (Cat# 101211, RRID: AB_312794), Ly6c-PerCP/Cy5.5 (Cat# 128012, RRID: AB_1659241), Ly6G-APC/Cy7 (Cat# 127624, RRID: AB_10640819), Gr1-PerCP/Cy5.5 (Cat# 108427, RRID: AB_893561), CD45-PE (Cat# 103106, RRID: AB_312971) and IFNγ-APC (Cat# 505809, RRID: AB_315403) (all obtained from Biolegend, San Diego, CA), and Granzyme B-PerCP-eFluor 710 (Cat# 46-8898–80, eBioscience, San Diego, CA; RRID: AB_11217678) were used.

## In vivo tumor growth and combination immunotherapy

CFP-B16 melanoma cells ($5 \times 10^5$) were subcutaneously implanted in the right flank of C57BL/6 mice (females, 6–8 weeks old). The tumor size was measured at five days post-implantation using a digital caliper. The volume of the tumor was calculated as V = L (length) × W (width) × H (height)/2 (*Huang et al., 2013*). Four days after the tumor inoculation, the mice in the CTX (Sigma-Aldrich, Saint Louis, MO) treatment group received an intraperitoneal (i.p.) injection of CTX diluted in sterile distilled H$_2$O at a concentration of 50 mg kg$^{-1}$, 100 mg kg$^{-1}$ and 150 mg kg$^{-1}$. Six days after tumor inoculation, the mice of the ACT and CTX-ACT groups were intravenously (i.v.) injected with $5 \times 10^6$ adoptive CTLs (250 µL total volume). The CTLs (five days after in vitro stimulation) were collected and prepared by centrifugation in Histopaque−1.083 (Sigma-Aldrich, Saint Louis, MO), washed three times and resuspended in ice-cold HBSS (HyClone). For the metronomic CTX-ACT treatment, the CTX dose schedule was 150 mg kg$^{-1}$ CTX injected i.p. on days 4, 9 and 14 after CFP-B16 tumor cell implantation followed by $5 \times 10^6$ CTLs (250 µL total volume) injected i.v. two days after CTX injection.

## Histologic analysis and immunofluorescence staining

Tumor tissues were fixed in 4% paraformaldehyde for 24–48 hr at 4°C, they were embedded in paraffin, sectioned, and stained with HE. For the immunohistochemical analysis, the sections were stained with markers of regulatory T cells (anti-Foxp3 antibody, 1:800, ab54501, Abcam, Cambridge, United Kingdom; RRID: AB_880110).

For the immunofluorescence analysis, tumor tissues were fixed in 4% paraformaldehyde for 24–48 hr at 4°C and then subjected to sequential dehydration in 10%, 20%, and 30% sucrose solution. The tissues were then frozen in OCT (Sakura, Torrance, CA) compound and cut into 20 µm slices. OCT was removed by three washes in PBS, and the tumor tissues containing EGFP cells were immunostained with Alexa Fluor 594 anti-mouse CD3 (1:150, Cat# 100240, RRID: AB_2563427), Alexa Fluor 647 anti-mouse CD4 (1:100, Cat# 100424, RRID:AB_389324), Alexa Fluor 647 anti-mouse CD8 (1:100, Cat# 100724, RRID: AB_389326), Alexa Fluor 700 anti-mouse Ly6G (1:200, Cat# 127622, RRID: AB_10643269), and Alexa Fluor 647 anti-mouse F4/80 (1:200, Cat# 123122, RRID: AB_893480) antibodies (all of which were obtained from Biolegend, San Diego, CA) to identify the T cells, neutrophils and macrophages, respectively. The tumor tissues containing YFP-DCs were immunostained with anti-MHC II (1:200, ab15630, Abcam, RRID: AB_302007) and anti-CD86 (1:100, ab25376, Abcam, RRID: AB_470491) and then incubated with mouse monoclonal anti-rat IgG2b/IgG 2a conjugated to Alexa Fluor 647 (1:2000, ab172333 and ab172335, Abcam, Cambridge, United Kingdom). CLSM was performed using a LSM710 (Carl Zeiss MicroImaging, Inc., Jena, Germany) with a 20× objective (N.A. 0.8).

## Preparation of the skin-fold window chamber and injection of CFP-B16 tumor cells

The window chambers were prepared as previously described (*Palmer et al., 2011*; *Schietinger et al., 2013*). Hair was removed from the back of the mouse one day before surgery. For the window chamber surgery, the mice were anesthetized by i.p. injecting a mix of ketamine (100 mg kg$^{-1}$, Sigma-Aldrich, Saint Louis, MO) and xylazine (10 mg kg$^{-1}$, Sigma-Aldrich, Saint Louis, MO) and positioned on a warmer plate at 37°C (Thermo Plate, TOKAI HIT, Shizuoka-ken, Japan). The skin on the back of the mice was sterilized with 70% alcohol and iodine solution. Titanium

window frames (APJ Trading Co., Inc., Ventura, CA) were implanted onto the back of the mouse, and then, a hole with a diameter of 1 cm was dissected by removing the skin and fascia on one side of the dorsal skin-fold flap while maintaining the integrity of the opposing dermis, fascial plane and vasculature. One day later, CFP-B16 tumor cells ($5 \times 10^5$ resuspended in 25 µL PBS) were injected at one site near the major vessel and between the fascia and dermis of the rear skin. The entire surgical process was conducted under sterile conditions to avoid infection. To relieve pain associated with surgery and inflammation, the mice received Tolfedine via i.p. injection (16.25 mg kg$^{-1}$, Vétoquinol, Québec, Canada) immediately and within 24 hr after implantation.

### Preparation of fluorescent dye labeled adoptive CTLs for intravital imaging

For intravital imaging, the adoptive CTLs were labeled with 25 µM CFSE (Invitrogen, Eugene, OR or 25 µM CellTracker Red CMTPX Dye (Invitrogen, Eugene, OR) for 15–45 min at 37°C as the standardized protocols described.

### Intravital imaging of the tumor microenvironment

The window-chamber mice were anesthetized by inhalation of 1.0–3.0% isoflurane in oxygen flow using a Matrx VMS small animal anesthesia machine (Midmark, Dayton, OH). The window was fixed on a warm plate (Thermo Plate) using a custom-made holder and then fastened to the microscope stage. Intravital images (*Qu et al., 2012*) were obtained using an A1R MP+ System (Nikon, Tokyo, Japan) with the large-field imaging function on a motorized stage. The images were captured using a 16× water immersion objective (N.A. 0.8) or 20× objective (N.A. 0.75, Nikon, Japan). Throughout the intravital imaging process, the temperature of the mice was maintained at 37°C with a warm plate. Using large-field imaging technology combined with blood vessel imaging as a 'position mark', the same imaging region could be focused on and images of the tumor could be obtained on different days. Confocal laser scanning microscopy (CLSM) was used to simultaneously image the CFP-B16 cells (405 nm laser, 400–500 nm emission), mRFP-Tregs and CMTPX-labeled adoptive CTLs (561 nm laser, 570–620 nm emission), CFSE-labeled adoptive CTLs and *Cxcr6*-GFP cells (488 nm laser, 500–550 nm emission). For the simultaneous imaging of the CFP-B16 cells and EGFP cells in vivo, multi-photon excitation microscopy was applied with an excitation wavelength of 860 nm.

### Data analysis

Intravital cell movement was tracked and analyzed with Image-Pro Plus (Media Cybernetics, Inc., Rockville, MD; RRID: SCR_007369) or Imaris 7.6 (Bitplane AG, Zurich, Switzerland; RRID: SCR_007370) software. The mean velocity, arrest coefficient, confinement ratio and mean displacement were calculated using Post-TrackObject software (custom-designed software) as previously described (*Sumen et al., 2004*; *Hugues et al., 2007*; *Cahalan and Parker, 2008*; *Matheu et al., 2011*; *Miller et al., 2002*). The arrest coefficient was calculated as the percentage of time that the instantaneous velocity of each cell was less than 2 µm min$^{-1}$ (*Boissonnas et al., 2007*), and the confinement ratio was calculated as the ratio of the maximum displacement of a given cell to its path length within a given time (*Cahalan and Parker, 2008*). Cells with a mean velocity of less than 2 µm min$^{-1}$ were defined as immotile (*Boissonnas et al., 2007*). The mean displacement plotted against the square root of time was calculated as previously described (*Ruocco et al., 2012*). Linear fitting was then performed on the plotted curves to determine whether the cells underwent random walking (*Cahalan and Parker, 2008*). An $R^2 > 0.95$ was evaluated as a good fit and the corresponding cell population was considered to be executing random walking.

### Statistical analysis

Statistical analysis was performed using GraphPad Prism 5 (GraphPad Software, Inc., La Jolla, CA; RRID: SCR_002798). For comparisons of three or more groups, the Kruskal-Wallis test was performed and followed by Dunn's multiple comparison tests (*Mrass et al., 2006*). For comparisons of two groups, the two-tailed unpaired *t*-test was performed. The statistical analysis is described in each figure legend.

## Acknowledgements

We thank Dr. Honglin Jin (Cancer Center, Union Hospital, Tongji Medical College, Huazhong University of Science and Technology, China), Dr. Wei Chen (University of Central Oklahoma) for paper discussion and Dr. Xiaohui Wu of Fudan University (China) for providing PB transposon system. We also thank Dr. Xiangning Li, Xiaohua Lv and Xiuli Liu of the Optical Bioimaging Core Facility of WNLO-HUST for the support in data acquisition. This work was supported by the Major Research plan of the National Natural Science Foundation of China (Grant No. 91442201), the National Basic Research Program of China (Grant No. 2011CB910401), Science Fund for Creative Research Groups of the National Natural Science Foundation of China (Grant No. 61421064), the Fundamental Research Funds for the Central Universities (HUST: 2015ZDTD014), the Director Fund of WNLO, and the 111 Project (No. B07038).

## Additional information

### Funding

| Funder | Grant reference number | Author |
| --- | --- | --- |
| National Natural Science Foundation of China | 91442201 | Shuhong Qi<br>Lisen Lu<br>Lei Liu<br>Zhihong Zhang |
| Ministry of Science and Technology of the People's Republic of China | 2011CB910401 | Shuhong Qi<br>Hui Li<br>Lei Liu<br>Qingming Luo<br>Zhihong Zhang |
| National Natural Science Foundation of China | 61421064 | Ling Fu<br>Qingming Luo<br>Zhihong Zhang |
| Ministry of Education of the People's Republic of China | 2015ZDTD014 | Shuhong Qi<br>Lisen Lu<br>Lei Liu<br>Zhihong Zhang |
| Ministry of Science and Technology of the People's Republic of China | Director fund of Wuhan National Laboratory for Optoelectronics | Shuhong Qi<br>Lisen Lu<br>Lei Liu<br>Qingming Luo<br>Zhihong Zhang |

The funders had no role in study design, data collection and interpretation, or the decision to submit the work for publication.

### Author contributions

SQ, Conception and design, Acquisition of data, Analysis and interpretation of data, Drafting or revising the article, Contributed unpublished essential data or reagents; HL, LLu, ZQ, LLi, LC, Acquisition of data, Analysis and interpretation of data, Drafting or revising the article; GS, LF, Analysis and interpretation of data, Drafting or revising the article, Contributed unpublished essential data or reagents; QL, ZZ, Conception and design, Analysis and interpretation of data, Drafting or revising the article, Contributed unpublished essential data or reagents

### Author ORCIDs

Zhihong Zhang, http://orcid.org/0000-0001-5227-8926

### Ethics

Animal experimentation: This study was performed in strict accordance with the recommendations in the Guide for the Care and Use of Laboratory Animals of Hubei Provincial Animal Care and Use Committee. The protocol was approved by the Animal Experimentation Ethics Committee of Huazhong University of Science and Technology (reference number: 452). All surgery was performed

under ketamine and xylazine, and all intravital imaging experiments were performed under 1-3% iso-flurane in oxygen, every effort was made to minimize suffering.

## Additional files

### Supplementary files

• Source code 1. The CellTracking_GUI.m is the main program of the Post-TrackObject software. The Post-TrackObject software is designed based on the Matlab (R2012, MathWorks) GUI.

• Source code 2. The CellTracking_GUI.fig is the main user interface of the Post-TrackObject software

• Source code 3. The CellTracking_GUI_Trajectory.m is the subprogram of the Post-TrackObject software

• Source code 4. The conrat.m is the subprogram of the Post-TrackObject software

• Source code 5. The dmean_motcoe.m is the subprogram of the Post-TrackObject software

• Source code 6. The mdmot_eb.m is the subprogram of the Post-TrackObject software

• Source code 7. The mean_sem.m is the subprogram of the Post-TrackObject software

• Source code 8. The trajectory.m is the subprogram of the Post-TrackObject software

• Source code 9. The vmean_arrcoe.m is the subprogram of the Post-TrackObject software

• Source code 10. The CellTracking_GUI_Trajectory.fig is the subinterface which corresponds to the subprogram CellTracking_GUI_Trajectory.m.

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
