## [Decision Letter]

Thank you for submitting your article "Long-term intravital imaging of the multicolor-coded tumor microenvironment during combination immunotherapy" for consideration by *eLife*. Your article has been reviewed by three peer reviewers, and the evaluation has been overseen by Xuetao Cao as the Reviewing Editor and Sean Morrison as the Senior Editor. The following individual involved in review of your submission has agreed to reveal their identity: Michael Hamblin (peer reviewer).

The reviewers have discussed the reviews with one another and the Reviewing Editor has drafted this decision to help you prepare a revised submission.

Summary:

By using intravital microscopy and a skin-fold window chamber models, the authors investigated spatio-temporal dynamics of immune cell subset infiltration and interaction in B16 melanoma-bearing mice receiving a combined chemoimmunotherapy of ACT plus CTX, and showed that Tregs could form an immunosuppressive ring around the CFP-B16 tumor that could be prevented by ACT and CTX. They also showed the dynamic process of the migration of adoptive CTLs into the tumor after selective depletion of Tregs by CTX treatment, and combined chemoimmunotherapy could also activate endogenous neutrophils and mature dendritic cells to infiltrate the tumor. Overall, the findings are interesting, contributing to better understanding of the improved efficacy of combined chemoimmunotherapy and providing translational potential in cancer therapy. However, some concerns should be addressed to improve the quality of this work.

Essential revisions:

The authors should select low-dose cyclophosphamide (50 mg/kg, but not 150 mg/kg) to repeat the in vivo therapeutic regimen and then observe the spatio-temporal dynamics of immune cell subset infiltration in B16 tumor. This would address multiple issues raised by multiple reviewers. What about the activation of endogenous CTLs by the combined immunotherapy, and how can you distinguish the infiltration of adoptively transferred CTLs and the endogenous CTLs? This also must be addressed.

Reviewer #1:

This paper uses in vivo fluorescence imaging of a multi-color coded fluorescent mouse tumor model to study immunotherapy with adoptive T cell transfer (ACT) and low dose cyclophosphamide (CTX). The main finding was that Tregs could form an immunosuppressive ring around the tumor that could be prevented by ACT and CTX. They also found that the treatment could activate endogenous neutrophils and mature dendritic cells to infiltrate the tumor.

1) They obtained the CTLs for adoptive transfer from mice that had been immunized using mitomycin C treated CFP-B16 cells. However, they have ignored a large body of work that treatments that produce damage-associated molecular patterns (DAMPs) in the tumor cells are more effective when used in vaccination protocols. DAMPs on the killed tumor cells are better able to activate dendritic cells.

2) As the authors will be aware there is considerable disagreement on the best dose for low-dose cyclophosphamide. Here they used 150 mg/kg which in my experience is quite high. In mice 50 mg/kg is more usual.

3) They have ignored several papers that have shown that fluorescent proteins (GFP) expressed in the tumor can act as tumor-associated antigens. How do they know that the CFP in the B16 tumor did not play a major role in the response?

4) They did not mention the activation of endogenous CTLs by the treatment. However, they did stain the tumors with anti-CD3 but it is not clear if they could distinguish between adoptive and endogenous T-cells.

5) If all the endogenous nucleated cells were stained with EGFP how could they distinguish between fibroblasts and endothelial cells and TIIs?

6) They have not mentioned the use of checkpoint inhibitors which is somewhat strange considering the enormous impact they have made in immune-oncology.

7) I think the Abstract needs more hard information; for instance about which cells were fluorescently labeled and something about the widow chamber and cell tracking. "immunosuppressive ring" is repeated twice.

Reviewer #2:

This manuscript by Dr. Zhang and colleagues focuses on spatio-temporal dynamics of immunocytes in B16 tumor bearing mice. Using intravital microscopy and a skin-fold window chamber model, the authors observed an "immunosuppressive ring" formed by Tregs surrounded the CFP-B16 tumor. This is an interesting paper addressing the dynamic process of the migration of adoptive CTLs into the tumor after selective depletion of Tregs by CTX treatment. Its principal finding is to demonstrate that the four stages migratory behavior of adoptive CTLs, including the mean velocity, confinement ratio and arrest coefficient, in the tumor microenvironment after CTX-ACT treatment. According to this finding, the authors designed a metronomic schedule of CTX-ACT treatment improving the anti-tumor efficacy of the combined immunotherapy, which has translational potential in cancer therapy. The comments are listed below for the authors' consideration.

1) In Figure 1, the authors isolated splenic and lymph node cells from B16 cell-vaccinated mice and cultured them in vitro with B16 lysates to amplify the cells for melanoma-specific CTL preparation. This method is rough. The authors should use CTLs from OT-I transgenic C57BL/6 mouse and OVA-B16 melanoma cells to further confirm the result.

2) In Figure 2, CTX is an immunosuppressive reagent and only low-dose CTX may function to selectively deplete Treg cells. The authors did not use low dose metronomic CTX but selected high dose CTX (150mg/kg) for the treatment. The authors should at least show the change of Treg cells and conventional T cells after 150 mg/kg CTX treatment.

3) In Figure 3, the authors used TII to instead TIL, which was weird.

4) In Figure 6, the authors emphasized the infiltration of DCs with mature phenotype. The data were not very strong.

5) In Figure 7, the authors designed the metronomic schedule of CTX-ACT treatment by 150 mg/kg CTX for treatment on day 4, day 9 and day 14. The author should provide more data to separate two possible consequences in order to explain the combined treatment. One was such dose CTX treatment only targeted Treg cells, thus relieving immune suppression. Another was such dose CTX resulted in much decrease of total T cell number, thus providing space for amplification of those adoptively transferred T cells.

6) Recognition of cognate antigen is essential for T cell to contact and attack tumor cell, however, chemokine is the most important factor regulating the T cells infiltration into tumor site. Please provide the rationale for why Treg cell depletion resulted in CTL accumulation.

7) Please provide more evidence to distinguish neutrophils and PMN-MDSCs.

8) In Figure 5—figure supplement 1B, the morphology of tumor cells between the PBS/ACT group and the CTX/CTX+ACT group is different. Was the scale bar different?

9) The English needs much improvement.

*Reviewer #3:*

This study by Qi et al. is very interesting! It revealed sequential anti-tumor immune responses, including adoptive cell therapy and host immune responses, in vivo in real time. The paper is clearly written.

Figure 2 shows no difference in tumor growth between CTX and CTX + ACT treated group. Dose this suggest CTL infiltrating tumors is not so important?

Figure 2: it would be helpful to quantify the Tregs and CTLs, as well as tumor "shrinking" and "melting", not only in CTX +ACT group but also other treatment groups. How many mice have been examined?

Does the number of infiltrating adoptive CTL inversely correlate with tumor growth?

Figure 4: It is would be very interesting to know the phenotype of the endogenous EGFP+ cells TIIs. Are there any CD8^+^ TILs or CD4^+^ Tregs? Co-staining will help.

Figure 5—figure supplement 1, Is there a significant difference in% neutrophil between the CTX and CTX-ACT treatment groups? It seems there is no difference, which suggests that neutrophil infiltration is the result of CTX. It seems there are not so many endogenous CD3+ T cells in the tumor. Does this mean that tumor elimination is mainly due to ACT?

In the subsection “Both CTX and ACT treatments are the essential factors for the chemotaxis and recruitment of TIIs into the tumor parenchyma”: "Among all of these groups, only the CTX-ACT treatment group resulted in the TIIs moving toward the tumor parenchyma…". It is hard to understand why only CTX-ACT, but neither CTX nor ACT treatment alone, induces this. Is it because those endogenous cells recognized something on the adoptively transferred CTLs or the transferred cells produced some factors to activate endogenous cells or promote their migration?

---

## [Author Response]

[…]

Essential revisions:

*The authors should select low-dose cyclophosphamide (50 mg/kg, but not 150 mg/kg) to repeat the* in vivo *therapeutic regimen and then observe the spatio-temporal dynamics of immune cell subset infiltration in B16 tumor. This would address multiple issues raised by multiple reviewers.*

Thank you for this valuable suggestion. The dose of cyclophosphamide (CTX) was the first thing we considered for designing the CTX-ACT combined treatment for CFP-B16 tumors. Previous reports used CTX doses between 30 and 200 mg kg^-1^ for the CTX-mediated Tregs suppression in mice (Ghiringhelli et al. 2004, Ercolini et al. 2005, Lutsiak et al. 2005, Motoyoshi et al. 2006). In different mouse tumor models, effective CTX doses between 50 and 150 mg kg^-1^ were applied for enhancing anti-tumor immune effects (Machiels et al. 2001, Le and Jaffee 2012), while a CTX dose range between 100 and 250 mg kg^-1^ was used in B16 melanoma chemoimmunotherapy (Sistigu et al. 2011).

In order to select a suitable CTX dose for CFP-B16 treatment, we determined the optimal dose experimentally. Therefore, we performed CFP-B16 tumor growth inhibition experiments with different CTX doses (50, 100, and 150 mg kg^-1^) on 4 days after 5 × 10^5^ CFP-B16 cells implanted into immuno-competent C57BL/6 mice and immuno-deficient BALB/c nude mice. These experiments confirmed that the anti-tumor effect of the CTX treatment with CTX doses between 50 and 150 mg kg^–1^ is not caused by the direct cytotoxicity of CTX, as no difference in the inhibition of tumor growth in immune-deficient mice (nude) was observed between the PBS control group and mice treated with different doses of CTX (50, 100, and 150 mg kg^-1^). In immune-competent mice (C57BL/6), the CFP-B16 tumor growth was significantly inhibited by treatment with 150 mg kg^–1^ CTX with or without adoptive T cells (ACT) in comparison to treatment with lower CTX doses of 50 and 100 mg kg^-1^ with or without ACT as well as PBS control groups. More importantly, CTX in lower doses (50 and 100 mg kg^-1^) with or without ACT failed to control the tumor growth and showed no significant difference to the PBS control group (Figure 2—figure supplement 1). These results suggest that the efficacy of the CTX treatment for CFP-B16 tumors depends on the enhanced anti-tumor immune response, but not on the direct cytotoxicity of CTX. According to the results of the CFP–B16 tumor growth inhibition experiments, we finally selected a CTX dose of 150 mg kg^-1^ for the CTX and CTX-ACT combined treatment.

To address this question, we added a new figure (Figure 2—figure supplement 1) and sentences in the Results to illustrate this question:

“This CTX dose was selected based on CFP-B16 tumor inhibition experiments with different CTX doses. […] Furthermore, by the tumor growth inhibition experiment performed on the BALB/c nude mice, we confirmed that the effective anti-tumor effect of CTX at the dose of 150 mg kg^-1^ is not due to its direct cytotoxicity (Figure 2—figure supplement 1).”

What about the activation of endogenous CTLs by the combined immunotherapy, and how can you distinguish the infiltration of adoptively transferred CTLs and the endogenous CTLs? This also must be addressed.

We designed two experiments to answer this question: the ex vivo characterization of endogenous T cells in the CFP-B16 tumors, and the intravital imaging of endogenous CTLs and adoptive CTLs in tumor areas on day 5 after the combined CTX-ACT treatment.

Here, we used transgenic mice (*Cxcr6^+/gfp^*) with the Cxcr6 sequence on one allele and the GFP replace CXCR6 coding region on the other allele, in which GFP cells with CD8^+^ marker represent endogenous CTLs (Unutmaz et al. 2000, Ruocco et al. 2012). We analyzed endogenous GFP cells in tumors of mice treated with PBS, ACT, CTX, and CTX-ACT on Day 5 (five days after ACT treatment and 11 days after implantation of CFP-B16 cells) by flow cytometry. This analysis showed that: (1) the number of endogenous GFP cells in the tumors of the CTX treatment group was significantly decreased with respect to the non-CTX treatment groups (1.31% versus 3.57%) and (2) most GFP cells (more than 60%) were CD8^+^ CTLs expressed with the activation marker CD69 in the tumors of CTX-ACT treated mice.

In order to distinguish adoptive and endogenous CTLs by intravital imaging, the adoptive CTLs were labeled with the red fluorescent dye CMTPX and transferred into *Cxcr6^+/gfp^* transgenic mice with CFP-B16 tumor cells implanted into the window chamber. The imaging data showed that the number of endogenous GFP T cells in the tumor areas of CTX treatment mice was much less than that in non-CTX treatment mice. The analysis of the dynamic behavior of the endogenous T cells showed that the mean velocity of the endogenous GFP T cells in CTX-ACT-treated mice was significantly decreased compared with ACT-treated mice (3.80 ± 2.56 μm min^-1^ in the CTX-ACT group versus 4.6 ± 2.0 μm min^-1^ in the ACT group).

Thus, both ex vivo characterization and intravital imaging suggest that the combined CTX–ACT treatment deleted most of the endogenous CD4^+^ T cells and retained the activated endogenous CTLs in the tumor area. The activated endogenous CTLs arrested in the tumor area and formed stable, long-lasting interactions with tumor cells to kill them efficiently.

To address this question, we added to the main manuscript a new section “CTX-ACT treatment decreased total endogenous T cells but retained activated endogenous CTLs in the tumor microenvironment” and two new figures (Figure 4 and Figure 4—figure supplement 1).

**“**CTX-ACT treatment decreased total endogenous T cells but retained activated endogenous CTLs in the tumor microenvironment

Next, we studied the dynamic behavior of endogenous CTLs in the tumor area. Here, we used *Cxcr6^+/gfp^* transgenic mice that carried the CXCR6 sequence on one allele and the GFP replace CXCR6 coding region on the other allele, in which GFP cells with the CD8^+^ marker represent endogenous CTLs (Unutmaz et al. 2000, Ruocco et al. 2012). […] Both ex vivo characterization experiments and intravital imaging suggested that the CTX-ACT combined treatment deleted most of the endogenous CD4^+^ T cells and retained the activated endogenous CTLs in the tumor area. The activated endogenous CTLs arrested in the tumor area for a long time and built stable, long-lasting interactions with the tumor cells to kill them efficiently.”

Reviewer #1:

[…]

1) They obtained the CTLs for adoptive transfer from mice that had been immunized using mitomycin C treated CFP-B16 cells. However, they have ignored a large body of work that treatments that produce damage-associated molecular patterns (DAMPs) in the tumor cells are more effective when used in vaccination protocols. DAMPs on the killed tumor cells are better able to activate dendritic cells.

We examined the released DAMPs in mitomycin C-treated CFP-B16 cells. The results demonstrated that although the mitomycin C treatment did not increase the release of DAMPs from CFP-B16 tumor cells, a small amount of DAMPs is released in the suspension of mitomycin C- treated CFP-B16 cells, which is beneficial for the activation of DC.

2) As the authors will be aware there is considerable disagreement on the best dose for low-dose cyclophosphamide. Here they used 150 mg/kg which in my experience is quite high. In mice 50 mg/kg is more usual.

Thank you for this suggestion. The selection of a CTX dose of 150 mg kg^-1^ instead of 50 mg kg^-1^ was based on literature values and the results of our tumor therapy experiments (details are shown in the “Essential Revisions 1”). The tumor therapy experiments confirmed that in comparison with other doses, the CTX dose of 150 mg kg^-1^ exhibited good efficacy to control the growth of CFP-B16 tumors, and the anti-tumor effect is due to the induced immune response but not related to direct cytotoxicity. Please see our response to Essential Revisions 1 for more details.

3) They have ignored several papers that have shown that fluorescent proteins (GFP) expressed in the tumor can act as tumor-associated antigens. How do they know that the CFP in the B16 tumor did not play a major role in the response?

In our study, the mitomycin C was used as the inhibitor of DNA proliferation of CFP-B16 tumor cells to get the whole-cell antigens to immunize the mice. To the best of our knowledge, mitomycin C does not induce the release of DAMPs from tumor cells when being used as cancer chemotherapy drugs (Tomasz 1995, Krysko et al. 2012). To answer this question, we performed the Western blot to detect the released DAMPs (including HMGB1 and Hsp70) in the supernatant and whole-cell lysates (as positive control) of CFP-B16 cells and B16 cells treated with or without mitomycin C (MMC). The results showed that both MMC-treated CFP-B16 and B16 tumor cells as well as non MMC-treated cells released a certain amount of DAMPs (Author response image). When C57BL/6 mice had been immunized with whole-cell suspension of MMC-treated CFP-B16 cells (2.5 × 10^6^ cells in 300 μL PBS), a certain amount of released DAMPs also promoted the activation of DCs (Krysko et al. 2012), which were beneficial for generating CFP-B16 specific CTLs.

Author response image 1.Western blot of DAMPs expression in supernatants or whole-cell lysates of mitomycin C treated B16 cells, non-mitomycin C-treated B16 cells, mitomycin C-treated CFP-B16 cells, and non-mitomycin C-treated CFP-B16 cells.Supernatants are normalized to the cell numbers.**DOI:**
http://dx.doi.org/10.7554/eLife.14756.069

4) They did not mention the activation of endogenous CTLs by the treatment. However, they did stain the tumors with anti-CD3 but it is not clear if they could distinguish between adoptive and endogenous T-cells.

Thank you for the suggestion. We have replied this question at “Essential revision 2”.

5) If all the endogenous nucleated cells were stained with EGFP how could they distinguish between fibroblasts and endothelial cells and TIIs?

For intravital imaging of the tumor microenvironment in EGFP-transgenic C57BL/6 mice, TIIs were identified from fibroblasts and endothelial cells according to their size, shape, and dynamic movement, as described in previous studies (Miller et al. 2002, Condeelis and Segall 2003, Mrass et al. 2006, McDonald et al. 2010, Yang et al. 2016). Furthermore, Furthermore, we identified the EGFP cells in situ by immunofluorescence histological analysis (Figure 6—figure supplement 2 A: CD3, Ly6G and F4/80), as described in the main manuscript.

6) They have not mentioned the use of checkpoint inhibitors which is somewhat strange considering the enormous impact they have made in immune-oncology.

Thank you for this good suggestion. The aim of this paper was to develop an excellent method for the in vivo detection of the spatio-temporal characteristics of various immune cells in the tumor microenvironment. This method is also suitable for monitoring the immune reaction of tumor-bearing mice treated with checkpoint inhibitors and comparing the efficacy between classical immunotherapy (such as the CTX-ACT combined treatment) and checkpoint inhibitors. In response to this concern, we added a sentence to the Discussion section:

“Furthermore, the method of long-term intravital imaging of various immune cells in the tumor microenvironmentin vivo is also suitable for monitoring multiple immune reactions, studying the spatio-temporal cellular events of other immunotherapies, such as checkpoint inhibitors (CTLA-4 antibody and PD-1/PD-L1 antibodies), and comparing the efficacy between classical immunotherapies (such as the CTX-ACT combined treatment) and checkpoint inhibitors on the treatments of different tumor models.”

7) I think the Abstract needs more hard information; for instance about which cells were fluorescently labeled and something about the widow chamber and cell tracking. "immunosuppressive ring" is repeated twice.

Thank you for the suggestion, we rewrote the Abstract:

“The combined-immunotherapy of adoptive cell therapy (ACT) and cyclophosphamide (CTX) is one of the most efficient treatments for melanoma patients. […] These insights into the spatio-temporal dynamics of immunocytes are beneficial for optimizing immunotherapy and provide new approaches for elucidating the mechanisms underlying the involvement of immunocytes in cancer immunotherapy.”

Reviewer #2:

This manuscript by Dr. Zhang and colleagues focuses on spatio-temporal dynamics of immunocytes in B16 tumor bearing mice. Using intravital microscopy and a skin-fold window chamber model, the authors observed an "immunosuppressive ring" formed by Tregs surrounded the CFP-B16 tumor. This is an interesting paper addressing the dynamic process of the migration of adoptive CTLs into the tumor after selective depletion of Tregs by CTX treatment. Its principal finding is to demonstrate that the four stages migratory behavior of adoptive CTLs, including the mean velocity, confinement ratio and arrest coefficient, in the tumor microenvironment after CTX-ACT treatment. According to this finding, the authors designed a metronomic schedule of CTX-ACT treatment improving the anti-tumor efficacy of the combined immunotherapy, which has translational potential in cancer therapy. The comments are listed below for the authors' consideration.

*1) In Figure 1, the authors isolated splenic and lymph node cells from B16 cell-vaccinated mice and cultured them* in vitro *with B16 lysates to amplify the cells for melanoma-specific CTL preparation. This method is rough. The authors should use CTLs from OT-I transgenic C57BL/6 mouse and OVA-B16 melanoma cells to further confirm the result.*

As mentioned in our manuscript, various reported studies have used the OT-I system with K^b^-OVA as a model antigen in T lymphoma cells and also in B16 melanoma cells. According to these reports, CTLs separated from OT-I transgenic C57BL/6 mice that displayed efficient tumor-killing abilities when they were transferred to mice bearing OVA-tumors (Kircher et al. 2003, Boissonnas et al. 2007, Shurin et al. 2009). Our unpublished results also confirmed that the adoptively transferred CTLs from OT-I mice completely eliminated the OVA-B16 tumors without any CTX treatment. However, unlike OVA are expressed in OVA-tumor cells, tumor antigens are usually expressed at a low level in most patients (Restifo, Dudley, and Rosenberg 2012). Thus, to mimic ACT immunotherapy in clinic, we chose an experimental system by using whole tumor cell antigens to prime and activate the CTLs. The method of generation B16-specific CTL refers to the book “Current protocols in Immunology” (Restifo 2011) and previous reports (Liu et al. 2006, Bauer et al. 2014). In response to this concern, we re-phrased the sentence in the Results as follows:

“To assess the curative efficacy of ACT on the murine melanoma *in vivo*, CTLs were obtained from the splenocytes of C57BL/6 mice immunized with mitomycin C pre-treated B16 melanoma cells according to previous protocols (Restifo 2011) and reports (Liu et al. 2006, Bauer et al. 2014).”

2) In Figure 2, CTX is an immunosuppressive reagent and only low-dose CTX may function to selectively deplete Treg cells. The authors did not use low dose metronomic CTX but selected high dose CTX (150mg/kg) for the treatment. The authors should at least show the change of Treg cells and conventional T cells after 150 mg/kg CTX treatment.

As the reviewer suggested, we detected CD4^+^ T cells, CD8^+^ T cells, and CD4^+^CD25^+^ Tregs in the blood of mice on days 5, 7, 11, 13, and 15 after implantation of 5 × 10^5^ CFP–B16 tumor cells. The mice were metronomically treated with 150 mg kg^-1^ CTX (intraperitoneally injected on days 4, 9, and 14) or PBS. The results showed that, compared with the PBS controls, the metronomic CTX treatments successively decreased the percentage of CD4^+^CD25^+^ Tregs in the blood of mice on days 11, 13, and 15 (1.44, 1.1, and 0.7% in the CTX group versus 2.28, 1.34, and 1.22% in the PBS group) and successively increased the percentage of CD4^+^ T and CD8^+^ T cells in the blood of mice on days 11 and 13 (29 and 31% in the CTX group versus 20.5 and 22% in the PBS group).

On day 15, CD4^+^ T cells, CD8^+^ T cells, and CD4^+^CD25^+^ Tregs were also examined in different organs (including spleens, TDLNs, and tumors) of mice treated with CTX or PBS. Compared with the PBS group, the CTX treatment decreased the percentage of CD4^+^ T cells, CD8^+^ T cells, and CD4^+^CD25^+^ Tregs in the spleen (all of them decreased about 50% in the CTX group compared with the PBS group), increased the percentage of CD4^+^ T cells (about 1.2-fold increase with respect to the PBS group) and CD8^+^ T cells (about 1.7-fold increase with respect to the PBS group) in the TDLN, and decreased the percentage of CD4^+^ T cells (about 30% decrease in the CTX group compared with the PBS group) and CD4^+^CD25^+^ Tregs (about 50% decrease in the CTX group compared with the PBS group) in the tumor. Notably, the CTX treatment had no effect on the percentage of CD8^+^ T cells in the tumor with respect to the PBS group (Figure 11).

Author response image 2.Alterations in T cell subsets caused by CTX treatment.Top row: percentage changes of CD4^+^ cells, CD8^+^ T cells, and CD4^+^CD25^+^ Tregs in the blood of normal mice, PBS-treated tumor-bearing mice, and CTX metronomically treated tumor-bearing mice on days 5, 7, 11, 13, and 15 after implantation of tumor cells. Bottom row: percentage changes of CD4^+^ cells, CD8^+^ T cells, and CD4^+^CD25^+^ Tregs in different organs (spleen, TDLN, and tumor) of normal mice, PBS-treated tumor-bearing mice, and CTX t metronomically treated tumor-bearing mice on day 15.**DOI:**
http://dx.doi.org/10.7554/eLife.14756.070

3) In Figure 3, the authors used TII to instead TIL, which was weird.

TII denotes tumor-infiltrating immunocyte, and TIL denotes tumor-infiltrating lymphocyte. In our experiments, more than 50% of the EGFP immune cells that infiltrated into the tumors were neutrophils and no lymphocytes (Figure 6—figure supplement 2). Therefore, we chose TII to indicate the EGFP-labeled tumor-infiltrating immunocyte instead of TIL.

4) In Figure 6, the authors emphasized the infiltration of DCs with mature phenotype. The data were not very strong.

Thank you for this valuable suggestion. We addressed this question by changing some words in the main manuscript to avoid any misunderstanding. We wanted to emphasize that the number of DCs infiltrating into the tumor area was significantly increased in the CTX-ACT combined treatment group, and some of the infiltrating DCs were mature. Here, we used MHC-II and CD86^+^ as the activated phenotype of DCs in the immunofluorescence assay, referring to the previous paper (Schiavoni et al. 2011).

5) In Figure 7, the authors designed the metronomic schedule of CTX-ACT treatment by 150 mg/kg CTX for treatment on day 4, day 9 and day 14. The author should provide more data to separate two possible consequences in order to explain the combined treatment. One was such dose CTX treatment only targeted Treg cells, thus relieving immune suppression. Another was such dose CTX resulted in much decrease of total T cell number, thus providing space for amplification of those adoptively transferred T cells.

Thank you for this good suggestion. Based on our previous data (Figure 1 and Figure 2) and new data (Figure 4, Figure 4—figure supplement 1 and Figure 11), we think both consequences are suitable for explaining the effective anti-tumor immune response of the CTX-ACT combined treatment. The combined treatment not only deleted most Tregs in the tumor areas to block the formation of immunosuppressive ring, but also provided some space for the amplification of adoptive CTLs in the tumor areas. In response to this suggestion, we rewrote the corresponding part in the Discussion section:

“Additionally, the Tregs depletion induced by CTX treatment contributed to the accumulation of adoptive CTLs in the tumor area by three ways: (1) it cleared the immunosuppressive barrier to facilitates the migration of adoptive CTLs into the tumor area; (2) it induced the high expression of some chemokines and cytokines in the tumor microenvironment to promote the accumulation of adoptive CTLs (Bracci et al. 2007, Schiavoni et al. 2011); and (3) it provided space to allow the infiltration and homeostatic proliferation of adoptive CTLs (Bracci et al. 2007, Sistigu et al. 2011).”

6) Recognition of cognate antigen is essential for T cell to contact and attack tumor cell, however, chemokine is the most important factor regulating the T cells infiltration into tumor site. Please provide the rationale for why Treg cell depletion resulted in CTL accumulation.

Thank you for this suggestion. We consider three possible reasons for the CTX treatment-induced Treg depletion, which results in the accumulation of CTLs in the tumor area.

Reason 1: As mentioned in our manuscript, the Tregs formed an “immunosuppressive ring” around the solid tumor, which blocked the ability of adoptive CTLs to recognize the cognate antigen of the tumor cells and decreased the number of adoptive CTLs migrating into the tumor area. The depletion of Tregs in the tumor area successfully broke this immunosuppressive ring so that adoptive CTLs efficiently penetrated into the tumor area without any barrier formed by immunosuppressive Tregs.

Reason 2: It has been reported that the Treg depletion due to CTX treatment resulted in the upregulated expression of some chemokines/chemokine receptors and cytokines (e.g., CXCL10/CXCR3 and IL–7) in the tumor area, which promoted CTL infiltration and homeostatic proliferation (Bracci et al. 2007, Schiavoni et al. 2011).

Reason 3: It has been reported that the proliferation of Tregs proceeds more vigorously than that of CTLs in the tumor area (Taylor, Neujahr, and Turka 2004, Sakaguchi et al. 2008, Sistigu et al. 2011). Thus, the CTX treatment induced the depletion of Tregs and the decrease of other endogenous T cells (Figure 4 and Figure 4—figure supplement 1) in the tumor area, probably providing space for the infiltration and homeostatic proliferation of adoptive CTLs (Bracci et al. 2007, Sistigu et al. 2011).

In response to this question, we added new sentences in the Discussion to illustrate this question. The additional sentence is presented below.

“Additionally, the Tregs depletion induced by CTX treatment contributed to the accumulation of adoptive CTLs in the tumor area by three ways: (1) it cleared the immunosuppressive barrier to facilitates the migration of adoptive CTLs into the tumor area; (2) it induced the high expression of some chemokines and cytokines in the tumor microenvironment to promote the accumulation of adoptive CTLs (Bracci et al. 2007, Schiavoni et al. 2011); and (3) it provided space to allow the infiltration and homeostatic proliferation of adoptive CTLs (Bracci et al. 2007, Sistigu et al. 2011).”

7) Please provide more evidence to distinguish neutrophils and PMN-MDSCs.

Neutrophils and PMN-MDSCs share the same markers and very hard to distinguish. T Previous reports demonstrated that in the mice the neutrophils and PMN-MDSCs are even functionally similar (Ramachandran et al. 2016). According to the previous papers, we characterized the neutrophils and MDSCs of EGFP cells ex vivo by flow cytometry (Figure 12). The markers of neutrophils are CD11b^+^Ly6G^+^Ly6C^low^, and markers of MDSCs are CD11b^+^Gr1^+^ (Ma and Greten 2016, Ramachandran et al. 2016).

Author response image 3.Characterization of the neutrophils and MDSCs of EGFP TIIs ex vivo by flow cytometry.The markers of neutrophils are CD11b^+^Ly6G^+^Ly6C^low^, and markers of MDSCs are CD11b^+^Gr1^+^.**DOI:**
http://dx.doi.org/10.7554/eLife.14756.071

8) In Figure 5—figure supplement 1B, the morphology of tumor cells between the PBS/ACT group and the CTX/CTX+ACT group is different. Was the scale bar different?

The scale bars are the same in the PBS, ACT, CTX, and CTX+ACT groups, which can be confirmed by the same morphology of the neutrophils in these four groups. The different morphologies of the tumor cells in the PBS/ACT and CTX/CTX+ACT groups is presumably caused by the anti-tumor immune response induced by the CTX treatment. The HE results showed that the anti-tumor immune response induced by the CTX treatment might affect the morphology of the CFP-B16 tumor cells.

9) The English needs much improvement.

Thank you for the suggestion, we have sent the manuscript to American Journal Experts for language editing to improve the quality of English writing.

Reviewer #3:

This study by Qi et al. is very interesting! It revealed sequential anti-tumor immune responses, including adoptive cell therapy and host immune responses, in vivo in real time. The paper is clearly written.

Figure 2 shows no difference in tumor growth between CTX and CTX + ACT treated group. Dose this suggest CTL infiltrating tumors is not so important?

Thank you for the suggestion, we think that the CTL infiltrating into the tumor area is very important for five reasons.

Reason 1: In CTX-ACT combined immunotherapy, it takes a long time to trigger the endogenous anti-tumor immune response. Although there was no significant difference in tumor growth control between CTX treatment and CTX-ACT combination treatment group, through intravital imaging, we found that the killing tumor effect were maximized, as well as the increased infiltration and motility of adoptive CTLs, in the CTX-ACT treated mice (Figure 2 and Figure 3).

Reason 2: According to our new data (Figure 4—figure supplement 1), the percentage of activated endogenous CD8^+^CD69^+^ CTLs in both tumors and TDLNs of the mice treated with CTX-ACT was higher than that of the mice treated with CTX alone (62% versus54% in the tumors, and 25% versus 19.3% in the TDLNs).

Reason 3: The intravital imaging data revealed that (Figure 6), compared with CTX treatment, only CTX-ACT treatment induced a rapid and direct movement of the endogenous TIIs toward the tumor parenchyma, which is beneficial for further anti-tumor immune response.

Reason 4: The intravital imaging data also confirmed that, compared with CTX treatment, only CTX-ACT combined treatment increased the amount of endogenous DCs infiltrating into the tumor area. Some of the infiltrating DCs were mature, which are beneficial for inducing the anti-tumor immune response.

Reason 5: The three rounds of the metronomic CTX-ACT combined treatment were designed according to intravital imaging information from a single round of the combined treatment. The results showed that the metronomic CTX-ACT treatment successfully controlled the growth of the tumor during its entire course, with a significant difference in the tumor growth control compared with the metronomic CTX treatments (P < 0.05, Figure 8).

For these five reasons, we considered that, besides the CTX treatment, the ACT treatment is also very important for controlling tumor growth. Combining the CTX with the ACT treatment gave synergistic effects on the anti-tumor immune response, which comprised in deleting most of the endogenous immunosuppressive Tregs, increasing the infiltration of adoptive CTLs and endogenous DCs, and accelerating the chemotactic movement of endogenous TIIs toward the tumor parenchyma.

Figure 2: it would be helpful to quantify the Tregs and CTLs, as well as tumor "shrinking" and "melting", not only in CTX +ACT group but also other treatment groups. How many mice have been examined?

Thank you for this suggestion. We imaged 3–4 mice (in two independent imaging experiments) in each group (including the ACT and CTX–ACT groups). To clarify this issue, we updated the figure legend of Figure 2. We also attempted to quantify Tregs and CTLs in different groups, but evaluated the quantification of Tregs and CTLs as being not sufficiently accurate. The number of Tregs in CTX-ACT treated mice was closed to zero, and the number of adoptive CTLs in mice treated with ACT alone was also close to zero. However, we added the results of this quantification in the Figure 2—figure supplement 2.

Does the number of infiltrating adoptive CTL inversely correlate with tumor growth?

Yes, the number of infiltrating adoptive CTLs correlates inversely with the tumor growth. According to the intravital large-field imaging and quantification data (Figure 1—figure supplement 3, Figure 2 and Figure 2—figure supplement 2), the number of infiltrating adoptive CTLs in tumor areas was higher in the CTX-ACT treatment group than in the ACT treatment group, which is consistent with the better tumor growth control in the CTX-ACT treatment group.

Figure 4: It is would be very interesting to know the phenotype of the endogenous EGFP+ cells TIIs. Are there any CD8^+^ TILs or CD4^+^ Tregs? Co-staining will help.

Thank you for this suggestion. According to our previous data, the phenotype of the endogenous EGFP TIIs includes T cells, neutrophils and tumor-associated macrophages (Figure 6—figure supplement 2A). The EGFP TIIs contained few CD8^+^ TILs and CD4^+^ T cells. We added the confocal imaging and immunofluorescence analysis of CD8^+^ TILs and CD4^+^ T cells in the Figure 13, which the frozen-sections of tumors were incubated with Alexa Fluor 647 anti-mouse CD4 antibody and Alexa Fluor 647 anti-mouse CD8 antibody to identify the endogenous EGFP CD8^+^ TILs and CD4^+^ T cells.

Author response image 4.Representative tumor sections stained with CD3, CD4, CD8, Ly6G and F4/80.Most of the EGFP TIIs at the tumor periphery were Ly6G^+^ and F4/80^+^. Scale bar: 50 μm.**DOI:**
http://dx.doi.org/10.7554/eLife.14756.072

Figure 5—figure supplement 1, Is there a significant difference in% neutrophil between the CTX and CTX-ACT treatment groups? It seems there is no difference, which suggests that neutrophil infiltration is the result of CTX. It seems there are not so many endogenous CD3+ T cells in the tumor. Does this mean that tumor elimination is mainly due to ACT?

Thank you for the suggestion. Here are the answers to the two questions:

1) The percentage of neutrophils in the CTX–ACT treatment group was slightly higher than in the CTX treatment group, but without significant difference. Although no significant difference was detected between the numbers of neutrophils in the tumor areas of CTX- and CTX-ACT treated mice, the mean velocity of neutrophils was significantly different (Figure 6). More importantly, only the CTX-ACT combined treatment elicited the chemotactic movement of neutrophils toward the tumor parenchyma (Figure 6). All these results confirmed that the efficient infiltration of neutrophils is induced by the synergistic effect of CTX and ACT.

2) Our immunofluorescence analysis of the tumor tissues showed only a small amount of endogenous CD3^+^ T cells in the tumor area (Figure 6—figure supplement 2A). According to our new data, the CTX-ACT treatment deleted most of endogenous T cells and only retained activated endogenous CD8^+^ CD69^+^ CTLs in the tumor areas (Figure 4 and Figure 4—figure supplement 1). These remaining activated endogenous CTLs also exhibited efficient tumor-killing abilities. Besides endogenous T cells, the activated endogenous neutrophils that infiltrated into the tumor area were also directly cytotoxic to the tumor cells (Piccard, Muschel, and Opdenakker 2012). As described in more detail in the answer to the first question of reviewer #3,the elimination of the tumor is due to the synergistic effects of the CTX-ACT combined treatment and failed upon the ACT treatment alone.

In the subsection “Both CTX and ACT treatments are the essential factors for the chemotaxis and recruitment of TIIs into the tumor parenchyma”: "Among all of these groups, only the CTX-ACT treatment group resulted in the TIIs moving toward the tumor parenchyma…". It is hard to understand why only CTX-ACT, but neither CTX nor ACT treatment alone, induces this. Is it because those endogenous cells recognized something on the adoptively transferred CTLs or the transferred cells produced some factors to activate endogenous cells or promote their migration?

Thank you for posing this stimulating question. According to the intravital imaging data (Figure 5,Figure 6), the synergistic effect of CTX and ACT treatments contributes to trigger the transiently increased chemotactic movement of TIIs toward the tumor parenchyma. The CTX treatment relieved the immunosuppressive microenvironment in the tumor, and promoted the expression of many cytokines (such as GM-CSF and IL-1β)(Bracci et al. 2007) to prime (Yao et al. 2015) the neutrophils and recruit the mature neutrophils into the tumor area. On the other hand, few adoptive CTLs infiltrated into the tumor area as early as 24 h after the ACT treatment; some adoptive CTLs killed tumor cells partially and also released some inflammatory cytokines to further promote the infiltration of endogenous neutrophils. According to the previous reports, the mature neutrophils exhibited high motility and efficient anti-tumor effects (Fridlender et al. 2009, Piccard, Muschel, and Opdenakker 2012, Sagiv et al. 2015). We think that this phenomenon observed by intravital imaging is worth further studies.